



# Mélange or landfast ice: What controls seasonal calving at Greenland outlet glaciers?

Sofie Hedetoft[1,2,*], Olivia Bang Brinck[1,2,*], Ruth Mottram[1,*], Andrea M. U. Gierisch[1], Steffen Malskær Olsen[1], Martin Olesen[1], Nicolaj Hansen[1], Anders Anker Bjørk[2], Erik Loebel[3,4], Anne Solgaard[5], and Peter Thejll[1]

[*]These authors contributed equally to this work.
[1]Department of National Centre for Climate Research (NCKF), Danish Meteorological Institute, Sankt Kjelds Plads 11, Copenhagen 2100, Denmark
[2]Department of Geosciences and Natural Resource Management, University of Copenhagen, Øster Voldgade 10, Copenhagen 1350, Denmark
[3]Technische Universität Dresden, Institut für Planetare Geodäsie, Helmholtzstraße 10, 01159 Dresden, Germany
[4]Alfred-Wegener-Institut Helmholtz Zentrum für Polar- und Meeresforschung, Sektion Glaziologie, Am Handelshafen 12, 48310 Bremerhaven, Germany
[5]Geological Survey of Denmark and Greenland, Øster Voldgade 10, Copenhagen, 1350 Denmark

**Correspondence:** Ruth Mottram (rum@dmi.dk)

**Abstract.**

Landfast sea ice and glacier mélange are part of a continuum of ice forms in front of marine-terminating outlet glaciers in Greenland. Mélange (*sikussaq*) has been posited to offer a buttressing effect on marine margins equivalent to floating ice shelves, potentially thereby helping to reduce the risk of marine ice sheet instability feedbacks. However, the role of mélange in

buttressing marine termini is controversial with previous studies showing mixed results and only limited effects on terminus ice velocities. Here, we use a comprehensive and novel in situ dataset of high time resolution GNSS position information, combined with satellite datasets of ice velocity and calving front position for three representative glaciers in north west Greenland. Our study at the Tracy, Melville, and Farquhar glaciers took place during the period from late winter (March) to peak melt season (July) in 2022 and 2023. Seasonal variations in outlet glacier velocity, calving activity and terminus position vary in-step with

the seasonal cycle of air temperature and landfast sea ice formation and break-up. Our observations are consistent with previous granular material theoretical frameworks where fast ice acts to delay the removal of mélange. However, we also observe large calving events at the peak of the fast ice season suggesting that neither landfast ice nor mélange fully suppress calving activity. We therefore suggest modelling landfast ice and glacier mélange as part of a glacier continuum that can modulate the response of glaciers to climate forcing on a seasonal cycle where landfast ice is seasonally present. The postulated buttressing or

backstress effect from the mélange appears mainly when it is bound by landfast sea ice, however we note that our observations show movements of the mélange away from the glacier fronts at a similar velocity, rendering the assumption that landfast ice or mélange exert a significant back stress on termini ambiguous. The break-up of landfast ice and onset of surface glacier melt occur concurrently in the summer melt season and both are probably therefore important in driving the seasonality of glacier front positions. We find no evidence of tidally driven movements within the mélange zone during the fast ice season, and no



effects from surface winds that may explain calving events. Our observations also form a comprehensive and useful dataset for evaluating models of mélange interactions and developing insights into the material properties of fast ice and mélange. We conclude that at these representative Greenland outlet glacier, landfast sea ice and not the presence of mélange controls the seasonal calving front behaviour.

## 1 Introduction

Ice mélange, the mixture of icebergs, calf ice and sea water in front of marine terminating glaciers, has been described as the world's largest granular material (Amundson et al., 2020). It forms a thick, and often impenetrable to humans, mass of calved icebergs at all size scales in front of calving outlet glaciers and is particularly prominent in Greenland and parts of Antarctica. It is a characteristic of many fast moving calving termini in Greenland, where it is seasonally glued together by landfast sea ice during the winter. Sometimes known as *sikussaq*, from an underdetermined etymology (see e.g. discussion in Fraser (2012)),

we here use the term mélange that has become more widely used in recent years in the glaciological literature. The disputed importance of mélange as a buttress to calving glacier termini has led to a great deal of recent interest in mélange processes and in the development of parameterisations of mélange buttressing in ocean and ice sheet models (e.g. Amundson et al. (2020); Wehrlé et al. (2023); Krug et al. (2015); Cassotto et al. (2021); Robel (2017); Vaňková (2021); Crawford et al. (2021)). However, there is little in situ data across seasons with sufficient temporal resolution and spatial coverage to test these models

and much of the literature is contradictory as to the importance of mélange on calving processes. Together with submarine melt, calving processes account for around half of the total ice mass loss from the Greenland ice sheet, with the remainder lost via melt and runoff of surface snow and ice (Otosaka et al., 2023; Shepherd et al., 2020). Calving and the melt of calved icebergs are also important controls on Greenland fjord circulation (Moon et al., 2018) and likely on the upwelling processes that drive biological productivity (Meire et al., 2017). However, the calving process itself remains poorly parameterised into

numerical models (Seroussi et al., 2020; Goelzer et al., 2020) and is a complex process with multiple proposed controlling and interacting factors such as glacier hydrology, meltwater plumes, ice mélange, ocean circulation, bed topography, and heat flux in both the atmosphere and the ocean (Benn et al., 2007). Inadequate representation in ice sheet models likely results in an underestimation of rates of sea level rise (Goelzer et al., 2020). In this paper we focus on the role of mélange and landfast sea ice in modulating calving rates in Greenland with a particular focus on the late winter, spring an early summer period when the

sea ice declines from its maximum extent and thickness in north west Greenland.

### 1.1 The Role of Ice Mélange in calving glaciers

The scientific community disagrees on the extent to which mélange directly impacts calving rates or vice versa (Robel, 2017; Krug et al., 2015; Crawford et al., 2021). Some studies have pointed towards the mélange as a mechanical inhibitor of the calving, where calving happens after a removal or substantial weakening of ice mélange, thus placing it as a controlling factor

(Amundson et al., 2010; Cassotto et al., 2021; Krug et al., 2015). Multiple studies (Krug et al., 2015; Bevan et al., 2019; Cook et al., 2021) suggest mélange can prevent crevasse propagation and thereby hinder and limit calving. It has also been suggested





that calving activity can influence the properties of the mélange through the generation of waves resulting in mélange dispersal and increased fracturing, thereby weakening the buttressing effect, potentially increasing further calving activities (Amundson et al., 2010). Joughin et al. (2020) suggest that ocean temperatures control the retreat and advance of marine-terminating outlet

glaciers in Greenland, via changes in mélange rigidity that helps to modulate the velocity of the mélange as it moves down the fjord, pushed by the outlet glacier. However, other studies underline that the buttressing effect of the mélange depends on characteristics in the mélange such as presence of landfast sea ice (Moon et al., 2015; Robel, 2017). Studies in Antarctica for example, (Ochwat et al., 2024; Fraser et al., 2021, 2023) suggest that mélange is less significant than thick multi-year sea ice in exerting a buttressing effect on outlet glaciers and can also explain seasonal calving dynamics (Greene et al., 2018). The

importance of mélange may also vary from glacier to glacier depending on the cohesion and density of the mélange (Wehrlé et al., 2023). Many studies of mélange in Greenland have focused on Jakobshavn Isbræ or Helheim glaciers. These are very large and constantly calving outlets of the ice sheet in long narrow fjords that may not be representative of other smaller and slower Greenland outlets where icebergs calve to form mélange. Similarly, analogies from Greenland may not be applicable at Antarctic ice fronts where, along with generally colder conditions that likely affect ice material properties, there is also a very

broad range of geometries ranging from narrow fjords (Surawy-Stepney et al., 2024), to embayments (Christie et al., 2022) and large open ice shelf fronts (Greene et al., 2018). It is thus important to consider local as well as general factors in studying ice mélange interactions with glaciers and landfast sea ice. In this study we combine high time resolution field observations and analysis of satellite data from Northwest Greenland to assess the combined influence of ice mélange and landfast sea ice on typical calving outlet glaciers. Based on field observations in the winter period we suggest defining an inner mélange zone

with very dense iceberg particles that can jam against each other (as in e.g. (Peters et al., 2015)) and an outer mélange zone with a lower concentration of icebergs. The inner zone is difficult to access and our field study is limited to the landfast sea ice and between the large icebergs of the outer zone. The distinction of zones is clear from satellite imagery from the summer open water period (see Fig. 1), but we note also that the mélange migrates considerably in the summer period and the distinction between inner and outer zones does not always hold.

## 1.2  Glaciology of Inglefield Fjord

Inglefield Bredning (Kangerlussuaq) is a wide gulf in Northwest Greenland with many glaciers discharging into the fjord that feeds it. Melville, Farquhar, and Tracy glaciers (Qeqertaarsuusarsuup Sermia, Toornaarsulissuup Sermii, and Tuttulipalluup Sermia in Greenlandic (Bjørk et al., 2015)) are located next to each other at around 77.6°N and 66.4°W, with their marine-terminating glacier fronts feeding into a common area at the head of Inglefield Fjord. The study area is visualised in Fig. 1.



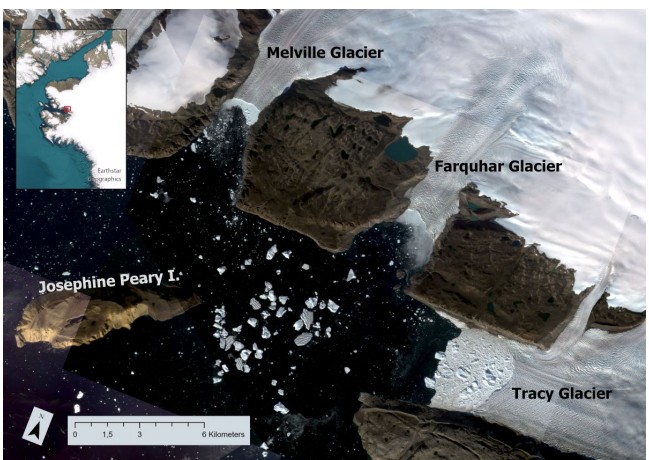

**Figure 1.** Map of Inglefield Fjord with the three study glaciers Tracy, Farquhar, and Melville marked as well as Josephine Peary Island (Qeqertarssussarssuaq). The basemap is a mosaic of optical satellite images between 25[th] and 29[th] of August 2022 obtained from Planet labs (planet.com/explorer)

Glaciers in the region have been well studied, particularly Tracy and Heilprin glaciers due to their contrasting recent retreat behaviour and to a lesser extent Melville and Farquhar glaciers have also been included in studies by e.g. Sakakibara and Sugiyama (2020); Willis et al. (2018). According to Sakakibara and Sugiyama (2018), Tracy glacier has recorded a consistent retreat since the 1980s and from 1980s-2014 it had the most rapid retreat of all glaciers in the area, with an average rate of $200 \pm 1$ m yr$^{-1}$, twice as fast as Farquhar ($94 \pm 3$ m yr$^{-1}$). The onset of Farquhar's retreat took place after the separation

of Farquhar and Tracy's shared glacier terminus in a floating ice tongue in the summer of 2002 (Sakakibara and Sugiyama, 2018). The geometry of the glacier fronts changed completely after the separation, as two separate tidewater glaciers were created, resulting in a notable change in the force balance of both Farquhar and Tracy. The new force balance combined with increasing meltwater enhancing basal sliding resulted in an ice velocity acceleration (Sakakibara and Sugiyama, 2018). Today, the ice mélanges of Farquhar and Tracy are also blended together in the winter season until the break-up of sea ice at the

head of Inglefield Fjord in July. Melville is the glacier with the lowest retreat rate of the three. It is located west of Farquhar and experienced a general retreat of $48 \pm 1$ m yr$^{-1}$ from 1980s-2014 (Sakakibara and Sugiyama, 2018). As with virtually all glaciers in Western Greenland, the glaciers of Inglefield Bredning are characterised by vertical thinning and retreat over the last decades. According to Wang et al. (2021), Tracy and Farquhar have the highest rates of elevation change in the fjord system at $-3.91 \pm 0.13$ m yr$^{-1}$ and $-2.91 \pm 0.18$ m yr$^{-1}$ from 2001-2018, respectively. The elevation change of Melville during the same

period was $-1.73 \pm 0.38$ m yr$^{-1}$, similar to the mean rate of outlet glaciers in Inglefield Bredning but larger than the average thinning at glaciers across Baffin Bay. Ocean observations by Willis et al. (2018) suggest that the rapid retreat of Tracy glacier is in part related to ocean warming and plume dynamics at the glacier front. In contrast, they suggest the lack of retreat and acceleration at neighbouring Heilprin glacier is most likely due to an outlet glacier front grounded in shallower water, though we note bathymetric data at all outlet glaciers ins not well constrained. The three study glaciers differ in size; Tracy is the largest





glacier with an area of 222.93 km$^2$, while Melville and Farquhar have areas of 118.52 km$^2$ and 53.74 km$^2$, respectively (Wang et al., 2021). In our study period, the measured width of the calving fronts in 2023 from satellite imagery was approximately 1.5 km for Melville, 2 km for Farquhar and 4 km for Tracy. Observations by Willis et al. (2018) confirm measurements reported in this study and in Rasmussen et al. (2021) of a warmer Atlantic layer situated below 200 m depth throughout the fjord system, consistent with other fjords in Western Greenland, indicating that turbulent mixing caused by subglacial plumes likely play a

key role in submarine melt at these glacier termini (Cowton et al., 2023; Slater et al., 2022). Unfortunately, the ocean depths in front of all three retreating glaciers in the study region are poorly known and mostly inferred from model inversions and limited radar observations (Willis et al., 2018; Morlighem et al., 2017). Bathymetric mapping of Inglefield Bredning using single beam echo sounding has been carried out as part of the Ocean Melting Greenland project Willis et al. (2018) but survey lines do not reach the melange zones of the Tracy glacier. The study glaciers represent a range of shallower and deeper grounded glaciers,

all of which can be seen to transiently float, leading to the occasional calving of tabular icebergs, observed in satellite imagery (see Fig. 1). These different types of outlet glaciers are representative for Greenland calving termini (Solgaard et al., 2022). Ice mélange is typically present throughout the year in this region and it's relative accessibility therefore provides an ideal process study site.

## 2 Methods and data

In this study we focus on the seasonal to annual cycle of calving, mélange and landfast sea ice. We focus in particular on the winter to early summer period when seasonal changes in fast ice provide a natural experiment to determine the role of mélange and landfast sea ice in buttressing glacier termini that is relevant to other calving glacier systems in Greenland. The winter landfast ice forms first at the head of the fjord likely aided by cold, fresh meltwater from outlet glaciers, advances from November through April towards the mouth of the fjord with maximum thickness and extent in April and May. From

June through to August break-up of fast ice is observed gradually in different regions of the fjord. Our aim is also to collect observational data that can be used for the development of ice sheet - ocean models that include mélange and sea ice processes. The area is readily accessible by dog sled in collaboration with local hunters and fishers enabling the installation and retrieval of field equipment over the course of multiple field visits.

### 2.1 In situ observations

#### 2.1.1 Mélange velocity from tracking buoys using GNSS

In order to monitor the movement of the mélange with higher time-frequency than would be possible by satellite as in other studies (e.g. Wehrlé et al. (2023); Bevan et al. (2019)), we deployed buoys in the mélange zone that recorded their position using GNSS at least every 30 minutes and transmitted this information using the Iridium communication network. We used two different buoy systems, TRUSTED and OMB in this study. TRUSTED buoys use a proprietary tracker system supplied

by Danish company Trusted A/S (trustedglobal.com), who designed and built the mounting of the GPS-tracker to a float





specifically for our use case. These devices are very robust but position recording settings are limited to few possibilities. OMB buoys, on the contrary, are open source instruments (Rabault et al. (2022) with details available at https://github.com/ jerabaul29/OpenMetBuoy-v2021a (accessed on 2024-07-10)). Hence these instruments are fully customisable and our OMB buoys were set to record their positions every 10-15 minutes. Pictures and more details about the two buoys systems, including

position uncertainty estimates are given in Table 1.

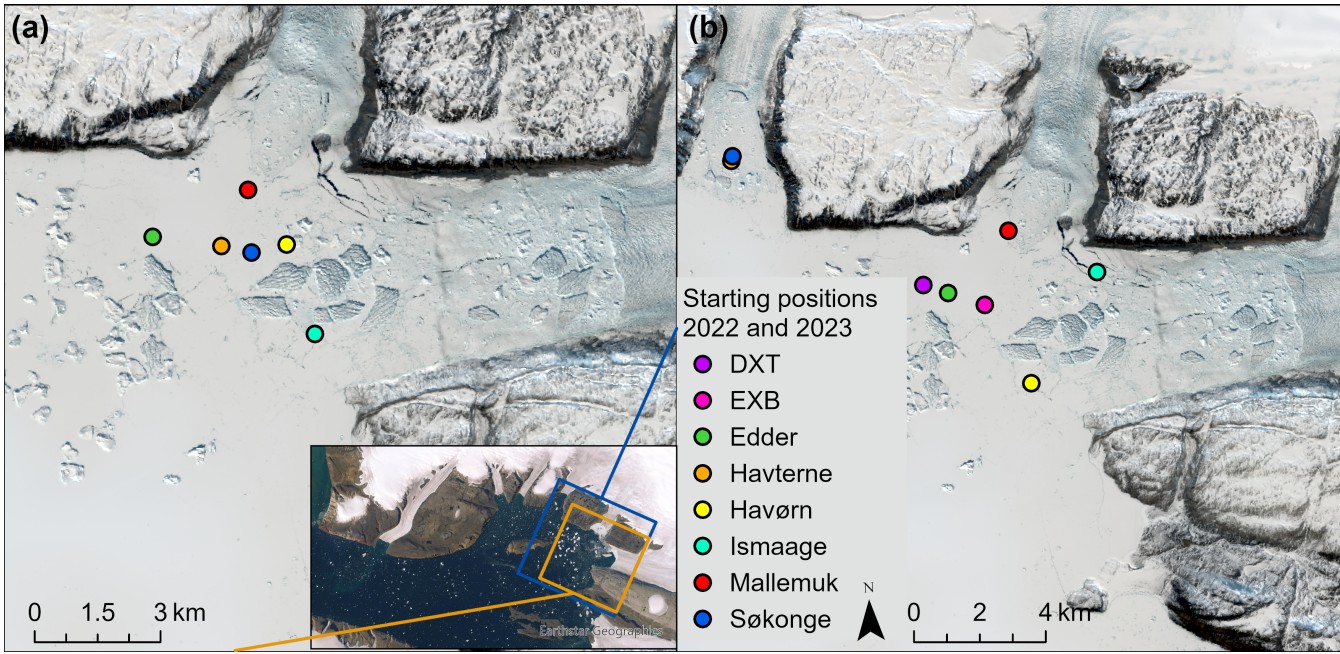

**Figure 2.** Starting positions of the buoys in 2022 (a) and 2023 (b). The basemap is Sentinel-2 from 25th May 2024, processed on Sentinel Hub EO Browser using a natural colour optimisation (Sinergise). Note the different spatial scales.

The TRUSTED buoys are a proprietary system that also record and transmit data including unit temperatures and battery life. Absolute coordinates are given in decimal degrees to 4 significant figures, which limits the position precision, but the buoys are robust and have a long battery life that allows them to be retrieved and redeployed over successive field seasons. The instrument is sealed in a metal box and attached to a float on a long metal rod that allows it to be deployed in a hole drilled into

sea ice and to float when the ice breaks up. The two OMBs were attached to inflated boat fenders and held upright by chains drilled through the ice. The plastic boxes containing the instruments were sealed with silicone prior to deployment. The buoys were deployed in the mélange zone at the end of March or in early April, when conditions allowed for travelling with dog sledges. When the landfast ice broke up in July, the floating buoys were recovered by boat by partners in the local community of Qaanaaq and could hence be re-deployed the year after. In March 2022, 6 buoys of type TRUSTED were deployed in

the mélange of Tracy and Farquhar glaciers only. In March 2023, 8 buoys (6 TRUSTED and 2 OMB) were deployed in the mélange of all three glaciers Tracy, Farquhar, and Melville. The deployment locations are shown in Fig. 2a for March 2022 and Fig. 2b for March 2023, respectively. The end of the study period is defined by the melt/break-up of the mélange. These



dates could easily be identified from the buoy position datasets as the buoy drift pattern changes noticeably when the buoys start to float in water. Hence, the buoy dataset covers the period of end of March to mid July. Velocity estimates of the mélange are derived from the recorded buoy positions. Due to uncertainty in the position data, especially of the TRUSTED buoys with limited accuracy (see Table 1), daily averaged positions of all buoys were used to calculate the daily mean velocity, which is the distance that buoys moved between two days (applying Geoprocessing tools in ArcGIS Pro). By investigating the buoy tracks on those days with especially high velocities, we additionally obtained the exact timing and total distance of several distinct abrupt buoy movement events and used these to identify specific drivers.

As an initial analysis suggested a possible diurnal cycle visible in OMB positions, and given the results of Cassotto et al. (2021), we carried out a spectral analysis of the velocity data to check if diurnal or tidal cycles were present. For this, instantaneous OMB-buoy velocities were deduced from the non-averaged position data available every 10 or 15 minutes. The resulting east–west and north–south components of the velocity were analysed for their dispersion. Additionally, power-spectra of the drift speed were calculated. As the velocity data are unevenly spaced and have gaps we used a Lomb-Scargle method (VanderPlas, 2018) to determine the power-spectra. A Hanning window was applied to taper the data to avoid spectral leakage and enable more accurate separation of signals.

### 2.1.2 CTD data

Ocean temperature and salinity profiles in the melange region of each glacier were measured during the deployment of the OMB and Trusted systems both in 2022 and 2023. We made use of newly calibrated Seabird SBE19plusV2 CTD with auxiliary sensors, deployed through a 12inch hole in the fast ice. Near full depth profiles were obtained and, together with analysis of a 14 year dataset assembled in the region, are currently in preparation for publication in a separate article.

## 2.2 Remote sensing data

We supplemented data on mélange deformation from in-situ data with remote sensing observations. These included both pre-processed ice velocity and calving front data, as well as novel datasets produced in this study. The purpose of the data comparison was to observe calving events, to estimate the location of the calving fronts, the size of the icebergs, as well as to identify transient clearing of ice mélange, the characteristics of ice mélange, the seasonal break-up of sea ice, the occurrence of melt water plumes at the surface and the movements and characteristics of large icebergs within the mélange zones (see Table A1)

### 2.2.1 Calving Dynamics from Earth Observations

We identified the calving events on the three glaciers by analysing the changing position of their calving fronts. For this, we used a dataset that was generated by a deep learning application on the multispectral Landsat satellite imagery (Loebel et al., 2024). The dataset defined 112 different calving front locations for Farquhar and Tracy and 81 for Melville in 2022 and 109 for all three glaciers in 2023.




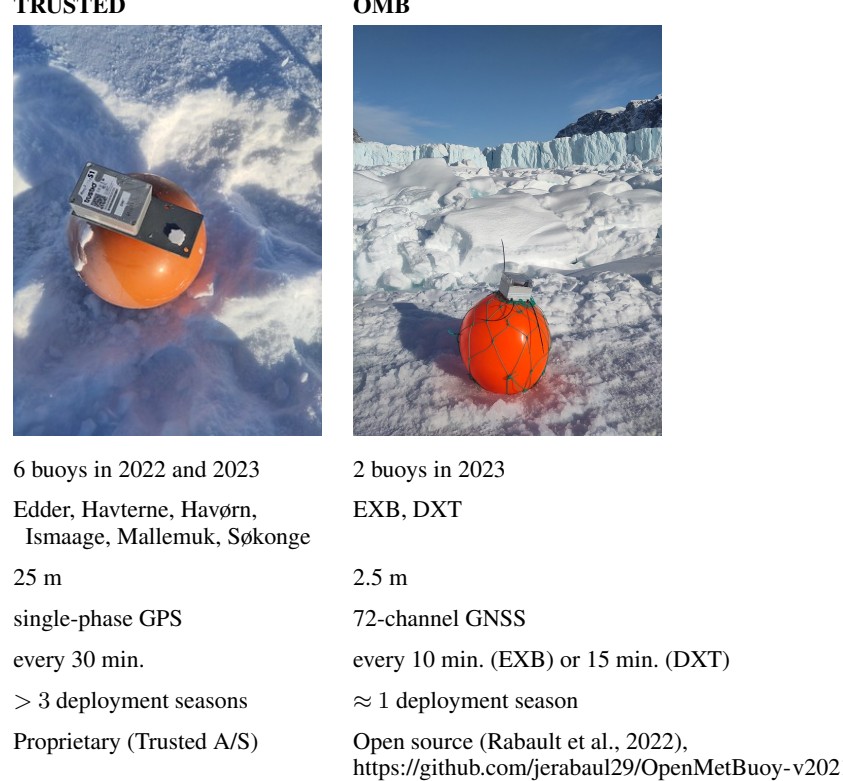

| | **TRUSTED** | **OMB** |
|---|---|---|
| Deployments | 6 buoys in 2022 and 2023 | 2 buoys in 2023 |
| Buoy names in this study | Edder, Havterne, Havørn, Ismaage, Mallemuk, Søkonge | EXB, DXT |
| Horizontal accuracy | 25 m | 2.5 m |
| GNSS system | single-phase GPS | 72-channel GNSS |
| Position frequency | every 30 min. | every 10 min. (EXB) or 15 min. (DXT) |
| Battery lifetime | > 3 deployment seasons | ≈ 1 deployment season |
| System design | Proprietary (Trusted A/S) | Open source (Rabault et al., 2022), https://github.com/jerabaul29/OpenMetBuoy-v2021a |
| Mounting | Pre-built buoy with metal pole | Laced to a fender with bottom weight |

**Table 1.** Specifications and pictures of the two buoy systems used in this study. The picture of the OMB is taken in the mélange in front of Melville glacier, with large tabular icebergs visible in the background.

In order to check the accuracy of the dataset and thus the analysis, we compared the deep learning-extracted calving fronts
with a manually delineated dataset consisting of 92 fronts on Tracy Glacier in 2022. Pre- and post-processing of the data
with the Feature Manipulation Engine (FME) were, along with the ArcGIS toolbox Glacier Terminus Tracking Tool (GTT)
(Urbanski, 2018), used to obtain an overview of days with large movements between two consecutive days indicating calving.
The fronts were manually delineated with the GEEDiT tool developed in Lea (2018) and the deviation between the two calving
front types were measured with a workbench created with FME (workbenches are archived for reproducibility see data section).
The median deviation across the front is 28 metres. This deviation aligns with the one originally calculated by Loebel et al.
(2024). However, in the context of day to day calving activity this imprecision is high. We also detected multiple days where the
deep learning-extracted calving fronts did not follow the correct calving front, but followed the position of the mélange instead
of the ice. We therefore supplemented our identification of calving dynamics calculated based on the automated calving fronts
with human assessed satellite observation to more precisely identify the daily changes. Despite the limitation of the optical
satellite on cloudy days without data coverage, close inspection of the satellite did allow us to identify the approximate time of




many large calving events in 2022 and 2023 for all three glaciers. The areal size indicated by the lost area of these events were calculated in ArcGIS Pro based on the automated calving fronts and daily Sentinel-2 imagery.

This key part of our analysis furthermore allowed us to connect satellite observed calving events with in situ buoy observations to understand the magnitude and timing of calving and mélange processes in the wider climate context of melt onset and the sea ice conditions.

### 2.2.2 Glacier and mélange velocity detected from satellite remote sensing

To assess the connection between outlet glacier velocity and mélange velocities, including the point measurements of in situ data, we used an ice velocity data product derived from the Sentinel-1 SAR data processed and published by PROMICE (Solgaard et al., 2021). The 20m resolution data product is continuously updated to give ice velocity with a 12–day gap across the Greenland Ice Sheet. The dataset is not masked to the ice sheet only, and thus includes areas of mélange at our study site. This enables us to compare satellite velocities with the in situ observed velocities. We selected two flowline points at higher and lower elevation on the three study glaciers as well as coordinates corresponding to the mélange in front of the glaciers, to compare time series of velocities derived by satellite with the in situ buoy measurements.

### 2.3 Regional climate and surface mass budget modelling

Previous work (Surawy-Stepney et al., 2024; Christie et al., 2022; Cassotto et al., 2015) link break-up and calving dynamics with seasonal atmospheric drivers, in particular thinning of sea ice and glacier surface melt and run-off driven by warming summer temperatures and wind forcing of mélange movements. We use two climate datasets to evaluate the importance of the surface boundary layer on mélange movements. The Copernicus Arctic Regional Reanalysis (CARRA) is a very high resolution (2.5 km) climate reanalysis that is updated in near real-time and has been shown to be particularly effective at capturing near-surface winds and sea ice (Køltzow et al., 2022). In this study we use the 6 hourly output wind field to assess if mélange velocities can be explained by persistent wind forcing. As CARRA does not include complex glacier surface processes, we also use surface melt and SMB driven by output from the HIRHAM5 regional climate model (RCM) (Langen et al., 2017). In this study, HIRHAM5 is forced with the ERA-5 (Hersbach et al., 2020) climate reanalysis data to identify the onset of melt and to quantify the amount of melt produced at the study glaciers. HIRHAM5 is run at a horizontal resolution of 0.05 x 0.05 degrees ( 5.5 km) on a rotated polar grid and has been shown to resolve the local climate well in this region (Lucas-Picher et al., 2012). The RCM outputs are further used to drive a detailed offline SMB model with 32 vertical levels (Langen et al., 2015, 2017), which, as analysis with in situ observations shows, reproduces observed firn processes like temperature and density well (Hansen et al., 2021; Vandecrux et al., 2024). Assessing uncertainties in climate, SMB and melt data is difficult in this region as we lack observations from low elevations, however a recent analysis of the simualtion by Puggaard et al. (2024) shows that HIRHAM performs well in both timing of melt onset as well as total number of melt days at higher elevations when compared with ASCAT satellite data. In this study we use the modelled outputs largely to assess timing of the onset of melt and runoff and the relative magnitude of melt events.



## 3  Results

### 3.1  Mélange Dynamics

The GNSS position observations show a consistent pattern of movement spatially and temporally in both 2022 and 2023 in Fig. 3 &  4, with the landfast sea ice and mélange moving slowly but continuously away from the glacier fronts, with occasional abrupt jumps in position recorded. The mélange velocities when bounded by landfast sea ice have a uniform and steady flow, but are slightly faster, particularly during abrupt movements, closer to the glacier fronts, and slower further away, with the direction of movement suggesting that the larger Tracy glacier has a greater influence on ice mélange velocity than the other

two termini. This would also indicate that Tracy is less sensitive to any buttressing effects from mélange and fast ice. The large abrupt extensional movements that intersperse the steady ice mélange drift are shown in Figs.  3 &  4. The buoys in the mélange have slightly different median velocities in the range of 7.6-11.3 m d $^{-1}$ in 2022 and 1.4-9.7 m d$^{-1}$ in 2023. The difference between the two field seasons is likely, at least in part, to be related to the different positions the buoys were placed in across the fjord, due to access differences between the seasons. Our results are indicative however, suggesting that proximity to

the glacier front as well as location relative to centre line of fast-flowing ice can explain differences in velocity, but a longer term and more spatially dispersed dataset is necessary to fully determine this. The abrupt movement events that we have identified in the record happen within the same 10-30 minute time intervals of all the GNSS buoys. It is therefore unfortunately not possible given the precision of the instruments to assess how quickly the signal travels between the buoys. We observe though, that in addition to happening concurrently, the measured speeds decrease with distance from the glacier. Our data indicates that the

largest of the measured abrupt movements represented up to 804 m of horizontal movement near Tracy Glacier during the half hour measurement period. These large abrupt movements likely represent large calving events at the glacier front. Interestingly, our analysis in Fig. 3 &  4 does not suggest that the velocity of the mélange steadily increases or decreases as time progresses and conditions change with melt and thinning sea ice in the later part of the winter season. We instead observe two distinct regimes, one where the mélange velocity is largely stable and there are a few abrupt movements, indicating sparse but definite

calving events in the winter period, and a second summer regime where the abrupt shifts in position become more common and of a higher magnitude. There is a break point where the melange shifts between these two regimes in mid-June in both years, suggesting that the onset of enhanced calving is related to the break up of landfast sea ice in the mélange zone, rather than the properties of the mélange itself.





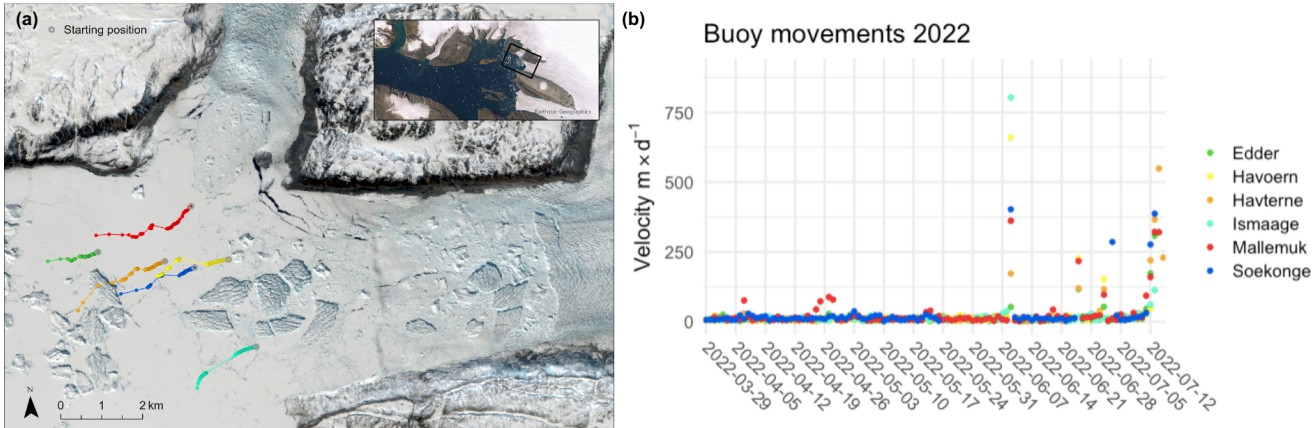

**Figure 3.** Movement of the TRUSTED buoys in the mélange of Tracy and Farquhar glaciers averaged over the day from the end of March until mid-July 2022. The basemap in (a) is an image from the Sentinel-2 satellite from 25th May 2024. The grey outline around the coloured points indicates the starting position.



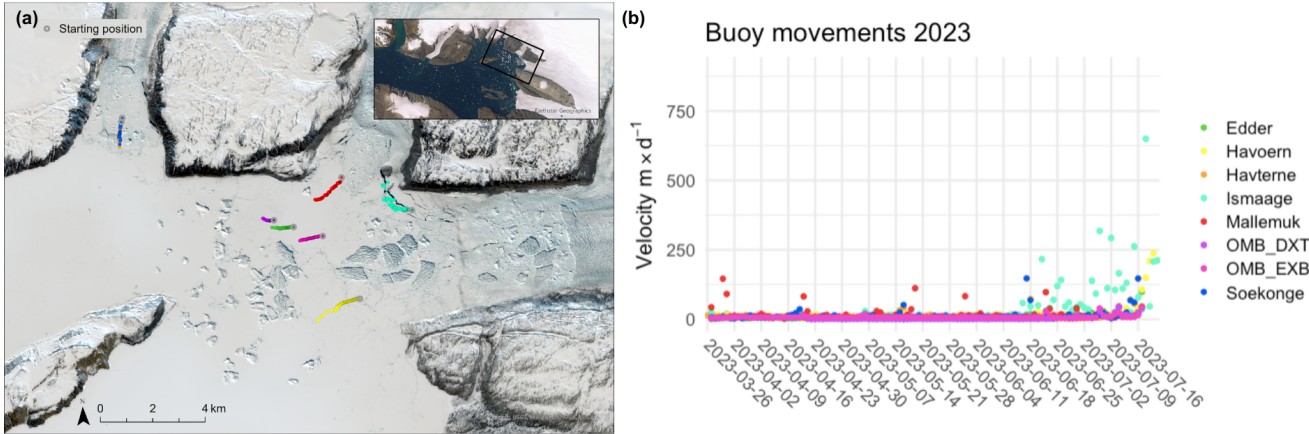

**Figure 4.** The movement of the 8 buoys (Edder, Havterne, Havørn, Ismaage, Mallemuk and Soekonge, DXT and EXB) placed in front of Tracy, Farquhar, and Melville daily averaged from the end of March until mid-July 2023. The basemap in (a) is Sentinel-2 from 25[th] May 2024 as in Fig 2. The grey outline around the coloured points indicates the starting position.

We summarise the observed calving events and distances moved by the specific buoys as a result in Table A1. It is difficult to give full uncertainty estimates on the position data as we do not have a differential GNSS system in operation at this site. We reduce noise introduced into the system by individual measurements by averaging all calculated velocities, derived from position information over each day. This smooths out noise introduced by GNSS signal variability. We note also that the abrupt jumps as well as the steady state drift of all buoys is consistent across all positions, suggesting that the stated position accuracy and precision of the buoys themselves is likely underestimated in Table 1. We have opted to omit the uncertainty bars on the figures in this paper for clarity, but full error estimates are available in the raw datafiles (see section on code and data availability). Buoys in front of Farquhar follow vectors that show an influence from Tracy as in Fig. 3 & 4 but are also being partly steered by the closer glacier. The buoys in front of Melville in 2023, show no influence from Tracy, only their local Melville glacier, but given the local topography and their position inside the local Melville glacier fjord arm, this is not surprising. We attribute the erratic back and forth movements of Ismaage (Fig. 4) in 2023 to its location on very thin sea ice over a bedrock sill in relatively shallow (150 m depth, measured in the field) water, where tidal mixing in the fjord below the ice causes regular melt and fracture of the ice on the corner of the fjord between Farquhar and Tracy glaciers (see satellite



timelapse video linked to in the video supplement section at the end). The in situ observed velocities are helpful to compare with and evaluate remote sensing derived glacier and mélange velocities (Fig. 5). The Sentinel-1 data has a 12 day repeat orbit, which means it cannot be used to assess the abrupt events identified in the GNSS buoy data. Our analysis in Fig. 5 shows the

265 GNSS velocities are similar to and centred around the satellite derived velocities of the lower part of Tracy glacier and the mélange in front of Tracy, with a tendency towards generally higher velocities in the GNSS observations of mélange. While the GNSS data have very similar average velocities to the satellite derived velocities, they have much higher daily variability, which it is not possible to capture given the image repeat time. Interestingly, our results confirm that the sea ice in the mélange zone and the outer fast ice zone is moving away from the glacier fronts slightly faster than the outlet glaciers are advancing,

even when the landfast ice is at its seasonal maximum thickness, indicating that neither mélange nor fast sea ice are exerting a strong resistive force on the glacier termini in this region.

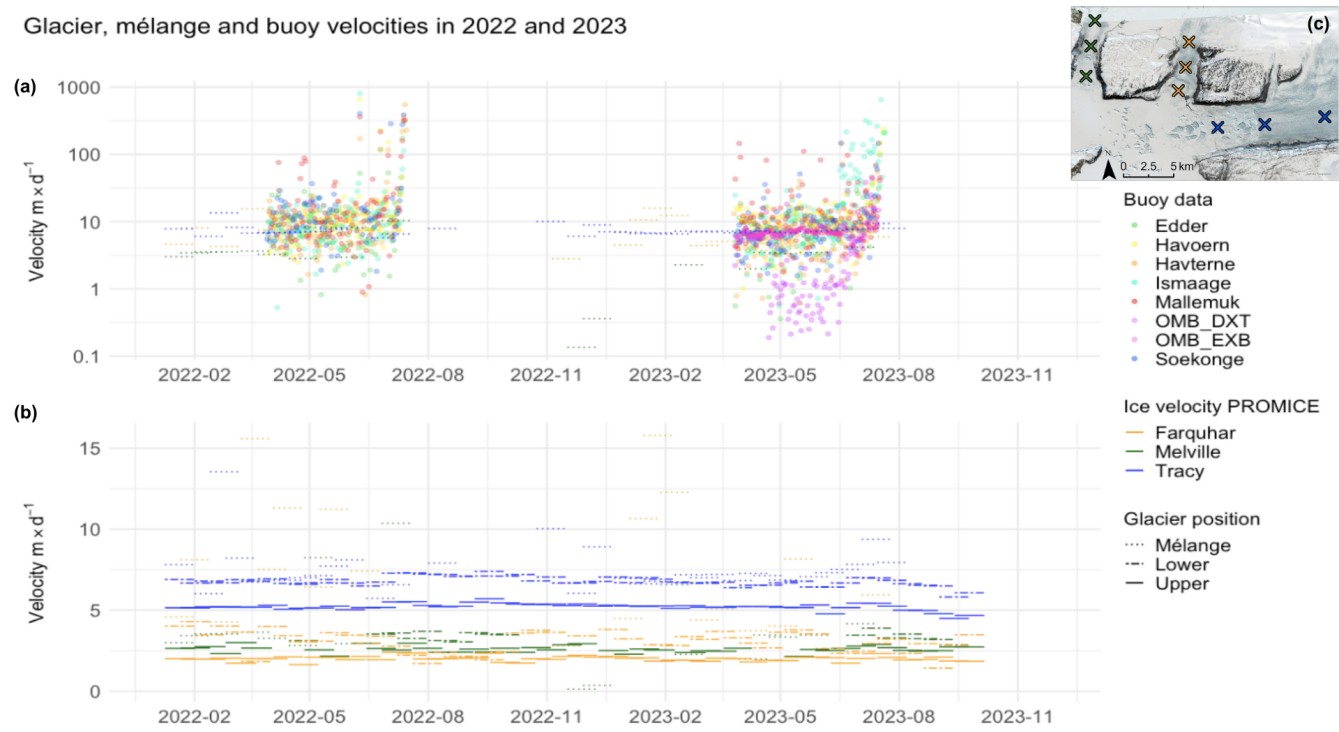

**Figure 5.** Mélange and glacier velocities for Farquhar, Melville and Tracy from the PROMICE Sentinel-1 ice velocity dataset (Solgaard et al., 2021) and in situ buoy movements. a) shows a comparison of the velocity of the mélange represented by in situ buoys (dots) and the PROMICE mélange product (dotted line), b) shows the velocity from PROMICE from the three positions shown on c) for the mélange, the upper and the lower point on each of the glaciers. Buoy marker colours in a) correspond to same colours as in Fig. 3 & 4 but are translucent to allow easier comparison with satellite derived velocities.





## 3.2 Calving front changes and outlet glacier dynamics

Glacier dynamics and calving activity at each of our three glaciers is similar to, but slightly varying from the characteristic seasonal pattern found by other studies for example, Black and Joughin (2023); Solgaard et al. (2022) at around 80 % of Greenlandic glaciers. Bézu and Bartholomaus (2024) identify 3 typical calving styles at Greenland outlet glaciers and while we observe all styles occurring at all three glaciers, we can also identify predominant styles at each glacier. Tabular rifting is a characteristic of both Melville and Tracy glacier, serac collapse is a common characteristic of calving at all glaciers but especially at Farquhar and slab capsize appears to be more common at Tracy glacier than the other termini. We therefore suggest that these glaciers are representative for a wider range of outlet glaciers in Greenland, though with the caveat that our study period only covers 2 winters and 3 glaciers. We see contrasting behaviour in the glacier front position at Tracy, Melville and Farquhar glaciers and these differences probably reflect the balance between long and short-term variability as well as contrasting calving styles. At Tracy (Fig. 6) and Melville glaciers (Fig. 8), there is a typical small winter re-advance of the terminus, though often also associated with calving activity, with a larger early summer advance followed by late summer and autumn retreat, often by a few large individual calving events. The full length of the terminus is shown in (Fig. 6) and the pattern of slow readvance and retreat is summarised in Fig 9. Farquhar glacier shows a similar but slightly less clear pattern in Fig. 7 with an obvious faster retreat on the western side of the calving front than on the east. This may be related to a series of small islands that can be seen appearing on the eastern side of the calving front and suggests the glacier is grounded in shallower water on that side of the front, which will influence the calving style (Benn et al., 2007). The crevasse patterns in Fig. 7 also suggest slower moving ice in this part of the glacier. Interestingly, the analysis of the automated calving front in Fig. 9b shows a general but rather fuzzier retreat pattern, possibly suggesting the deep learning algorithm found it harder to map the calving front at Farquhar glacier.



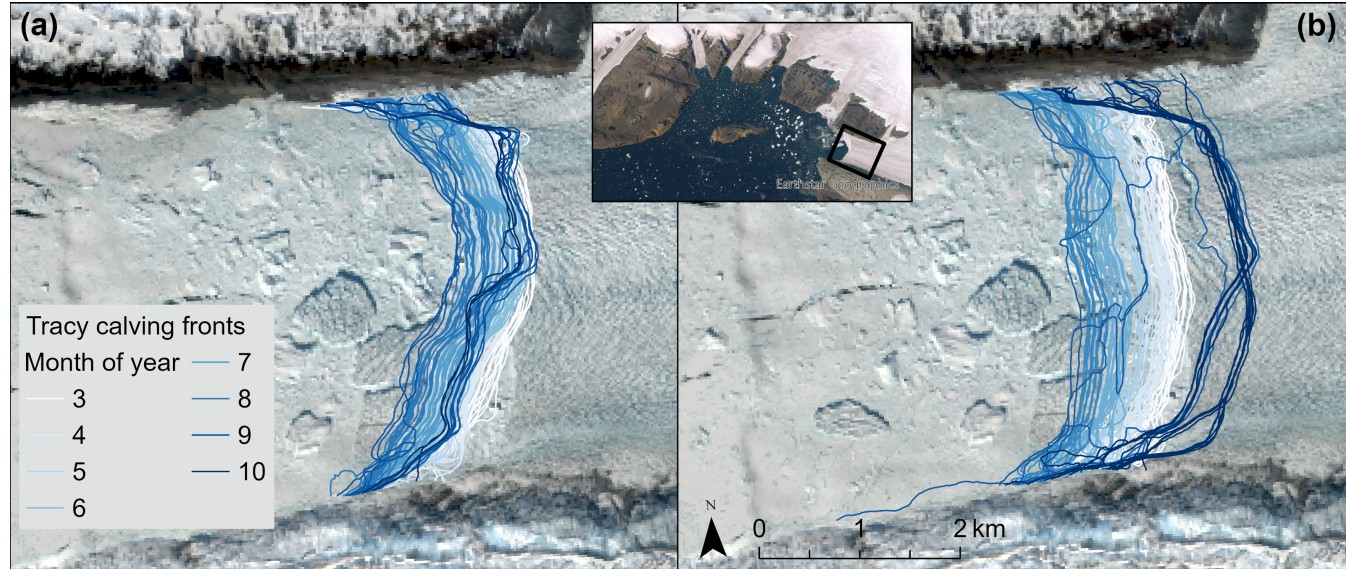

**Figure 6.** Calving front positions for Tracy Glacier in 2022 (a) and 2023 (b) generated with deep learning (Loebel et al., 2023). The basemap is Sentinel-2 from 25$^{th}$ May 2024, processed on Sentinel Hub EO Browser using a natural colour optimisation (Sinergise). Note the different spatial scales.



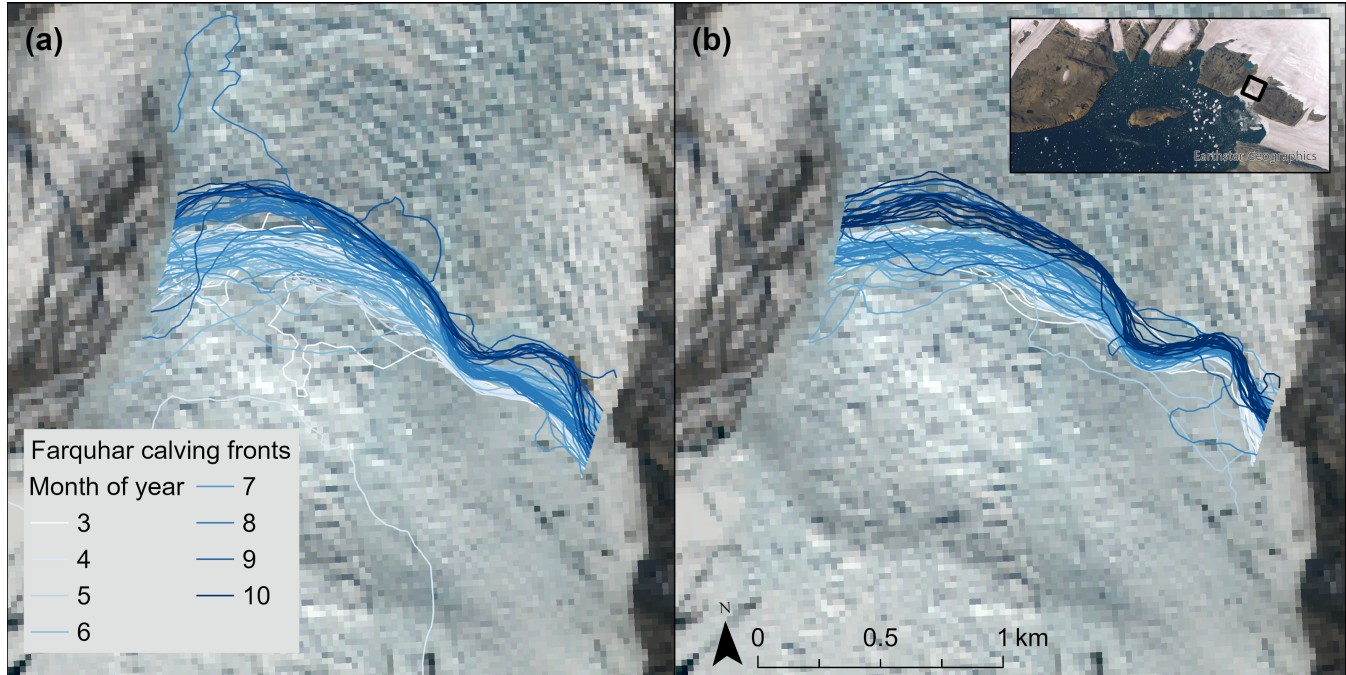

**Figure 7.** Calving front positions for Farquhar Glacier in 2022 (a) and 2023 (b) generated with deep learning (Loebel et al., 2023). The basemap is Sentinel-2 from 25$^{th}$ May 2024, processed on Sentinel Hub EO Browser using a natural colour optimisation (Sinergise). Note the different spatial scales.




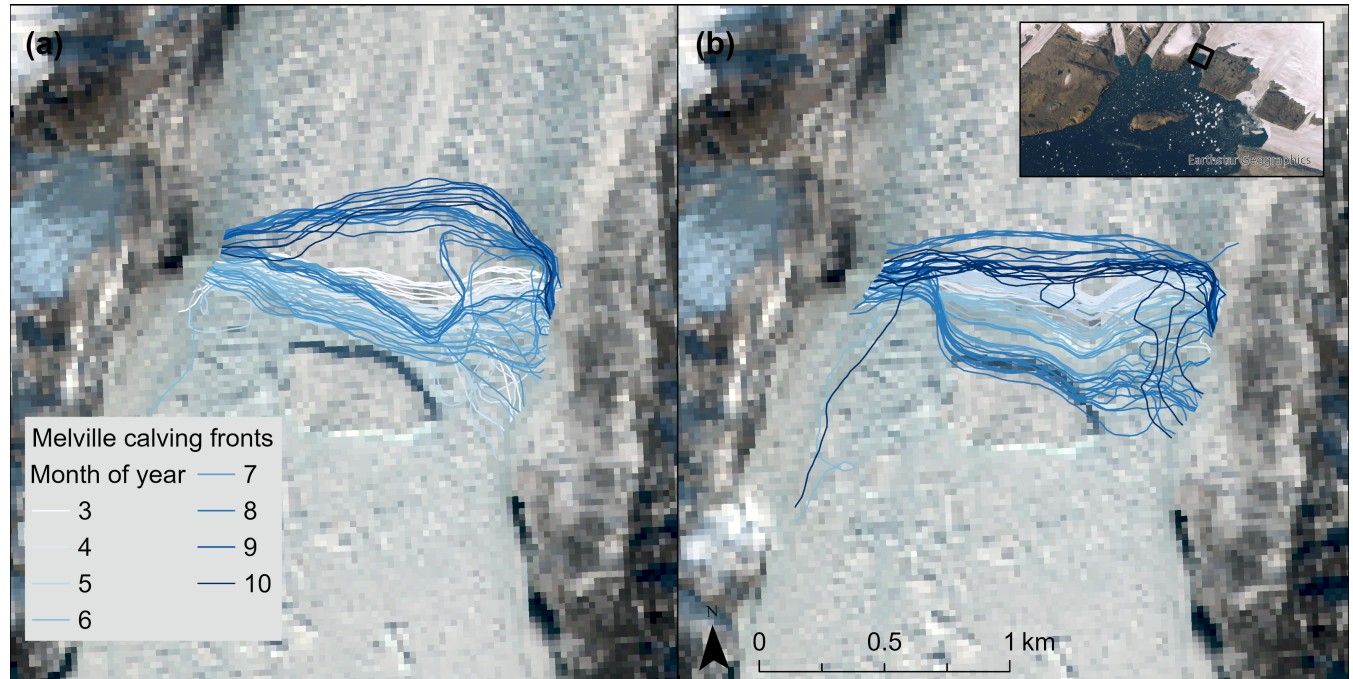

**Figure 8.** Calving front positions for Melville Glacier in 2022 (a) and 2023 (b) generated with deep learning (Loebel et al., 2024). The basemap is Sentinel-2 from 25th May 2024, processed on Sentinel Hub EO Browser using a natural colour optimisation (Sinergise). Note the different spatial scales.




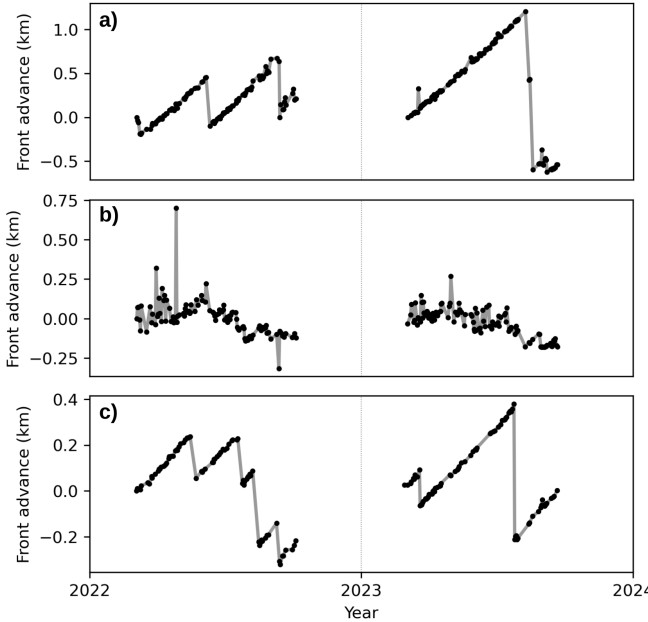

**Figure 9.** Calving front time series at the central flow line at (a) Tracy glacier, (b) Farquhar glacier and (c) Melville glacier for 2022 and 2023. Calving front positions are marked with black dots, and solid gray lines connect entries for each year (Loebel et al., 2023).

With the combination of the automated-extracted calving fronts (Loebel et al., 2024), buoy data showing abrupt velocity changes, and Sentinel-2 satellite imagery, we manually identified different calving events happening in total on 40 days from March–September in 2022 and 2023. Some events may have occurred over multiple days but are noted as one event happening
on the same date due to the available temporal and spatial resolution of satellite imagery. These calving events and changes in velocity are summarised in Fig. 10. We analysed correlating factors for 21 of the 40 identified events and these are summarised in the appendix Table A1. The measured sizes of the icebergs produced in these calving events are also indicated. This effort allows us to closely connect changes in mélange and landfast ice with calving events.

There are five large calving events on Tracy in 2022 and 2023. The events detected on Tracy happen from June to September
and are larger than the events detected at Melville and Farquhar. Events at Melville Glacier were detected from March to the end of September. Melville is the glacier with most events both in 2022 and 2023 (10 in 2022 and 14 in 2023). Farquhar produces the smallest calving events on average, and we detected four in 2022 and seven in 2023. The majority of the events happen in the summer when there are positive daily air temperatures, but all three glaciers show calving events well outside of the melt season, including during the period with thickest sea ice and mélange. This conclusion is supported by both the
satellite imagery and in situ buoy data. Tracy's front for example retreated almost a kilometre on the northern side during the winter of 2022-2023 indicating winter calving (Fig. 6). A similar 1 km retreat is detectable at Melville glacier between 2022 and 2023 (Fig. 8). 11 out of 15 events detected in 2022 happen after sea ice break-up, while the calving is more equally distributed in 2023, with 14 events after sea ice break-up and 10 before, of which 3 occurred around the identified break-up





date. We note that smaller calving events, which in fact likely make up the majority of all calving events, are undetectable by
310 the manual methods chosen here but are integrated into the calving front changes.

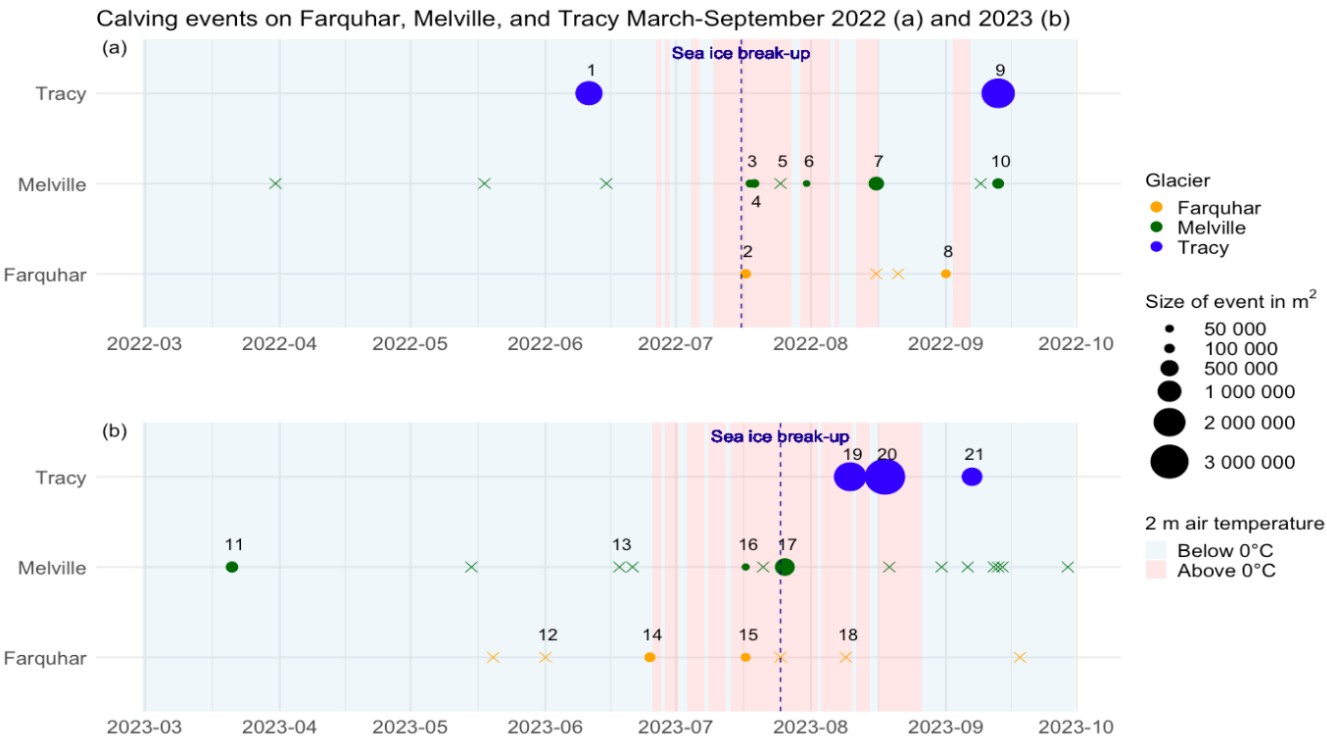

**Figure 10.** All identified calving events on Melville, Farquhar and Tracy Glacier in 2022 (a) and 2023 (b). An event is plotted on the first detectable day (in satellite imagery) after the event has happened on the x-axis and distributed to its glacier location on the y-axis. Further, the colours orange, green and blue represent Farquhar, Melville, and Tracy. The approximate spatial size of each calving event in $m^2$ (if measured) are indicated by the size of the points. If it is not possible to determine the size of the calving event, it is marked as a cross. Red and blue backgrounds symbolise positive and negative daytime air temperatures, respectively, based on 2 m air temperature data from CARRA at 12:00 PM UTC. The numbers (1-21) added to some points are reference numbers for events given in Table A1 and the dashed vertical lines represent sea ice break-up dates: 16/7 2022 and 25/7 2023, identified from changes in buoy movements and satellite imagery.

### 3.3 Atmosphere and Ocean Processes

Glacier mélange is influenced by ocean and atmospheric processes. We therefore include observations ocean processes and state as well as information from atmospheric reanalysis in our study.





### 3.3.1 Ocean and Sea Ice Conditions



**Figure 11.** Vertical profiles down to 400m depth of temperature (red) and salinity (black) in front of Tracy glacier in March 2022 (thick lines) and March 2023 (thin lines) Also shown is a March 2022 temperature profile at the mouth of the fjord at the edge of the landfast sea-ice (dotted).

In Fig. 11 we show two CTD profiles from similar locations in the centre of the mélange area between the Tracy and Farquhar glaciers from 2022 and 2023. Overall we note little difference between profiles from 2022 and 2023, though there is a larger difference with observations from the mouth of the fjord around 100km to the west, near the edge of the sea ice. The glacier profiles reveal a cold winter upper mixed layer reaching only 10-15m above warmer layers reflecting upwelling and modifica-





tion of warm deep water of Atlantic origin. In the profiles off Tracy glacier, the temperature reaches a maximum 0.2 to 0.3°C
at bottom levels. Below the mixed layer salinity gradually increases leaving the water column well stratified. Atlantic water
core that exceed 1 ° C at depths of around 300-350 m are found at the mouth of the fjord, consistent with previous observations
(Jackson et al., 2021; Willis et al., 2018). In contrast to the shallow mixed layer in the melange region off Tracy, signatures of a
+100m deep winter mixed layer is seen in the profiles at the mouth of the fjord. Future work will focus on defining the typical
range of iceberg sizes and thickness in the mélange, based on field observations, however in the field in this study we estimated
large tabular icebergs near the GNSS buoys to be in the 10 to 20 m height above water line range, to a maximum of 30 m. Most
of the icebergs in the mélange zone are considerably smaller than the few exceptionally large tabular icebergs, and the landfast
sea ice in between was consistently measured to be around $1 \pm 0.2$ m thick. A 30 m high iceberg may reach depths of about
300 m and therefore into the warm Atlantic water layer that could potentially assist in accelerating melt, but the vast majority
of the mélange is sitting in the cold and fresh upper layers. We therefore interpret the temperature and salinity in the deep water
as being unimportant for the changes in mélange and especially sea ice thickness during the seasonal cycle. Thinning and melt
of the winter ice in summer, and well as formation and growth in the late autumn and winter is largely driven by atmospheric
processes.

### 3.3.2    Ocean Tides

Initial analysis of the first EXB OMB data in 2023 showed a potentially quasi-diurnal cycle, though no cycle was visible in
the DXT data, in spite of its nearby location. As other studies (Cassotto et al., 2021) have suggested that there may be a tidal
forcing on some mélange zones and associated calving, we carried out further analysis to try to distinguish if there are diurnal
and/or tidal processes visible in the mélange movements. Spectral analysis of the high time resolution OMB data does not show
a tidal signal, Fig.  12 &  13 show the velocity dispersion for each buoy and the power-spectra, respectively. We see that for the
buoy EXB the north-south velocities are clearly larger than the east-west velocities, suggesting more cross-fjord motion than
along-fjord. DXT does not appear to behave like this, showing rather equal velocity dispersions cross- and along-fjord.
     Several signals at a level considered statistically significant are present in the power spectra in Fig. 13, such as a 24-hour
signal and several shorter periodicities. Notably absent are any periods that could be related to tides such as the principal lunar
semidiurnal periodicity, M2 (12 hours 25 minutes) which is entirely lacking. The 24-hour signal is present in both the first and
last half of the data records for both buoys, although not of equal strength over time. The presence of apparent diurnal effects
in GNSS has been well documented (Weiss, 1989) and a similar decomposition of data from the GNSS sensor in Copenhagen
showed similar features. We therefore conclude that unlike effects reported by Walter et al. (2012) and Cassotto et al. (2021),
tidal effects are not significant at these glaciers in affecting mélange velocity at least when the fast ice present. Our results here
are similarly a cautionary tale for groups attempting to replicate this work at other glacier termini.

### 3.3.3    Winds

As wind stress can also influence the transport of icebergs and mélange, we analysed the dominant wind directions to assess
if buoy positions were likely to be responding to surface winds and in particular if the across fjord movements of EXB could





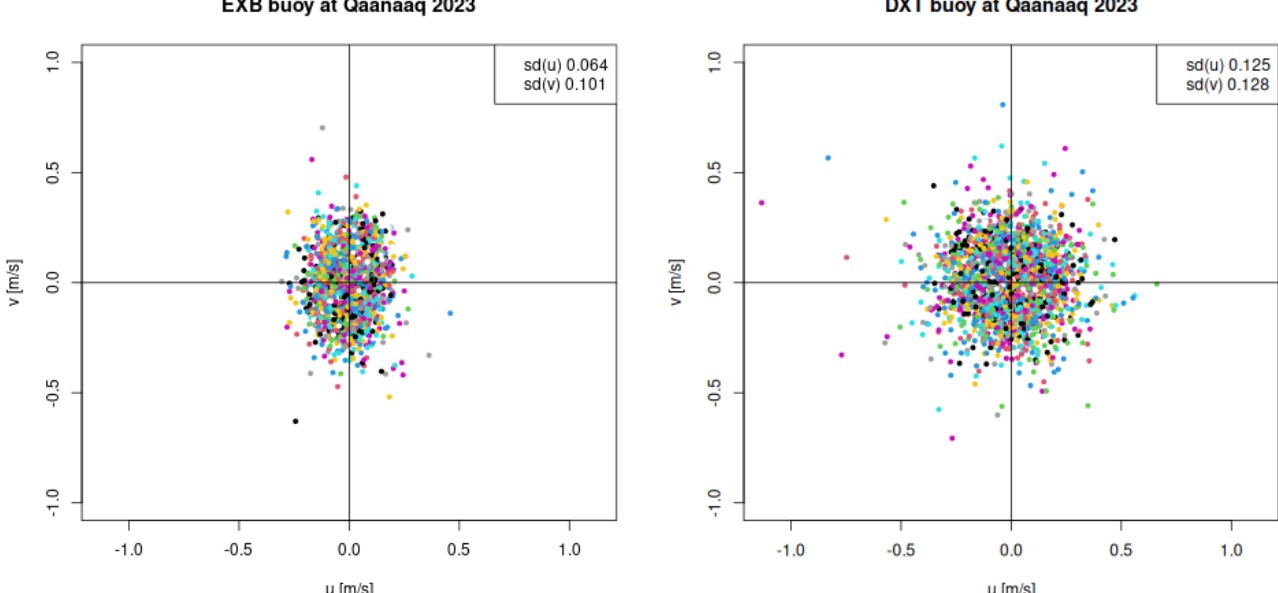

**Figure 12.** East-west (u) and north-south (v) velocity components for the high resolution OMB buoys EXB (left) and DXT (right) buoys, based on GPS positions and time-stamps. Colours correspond to the different timestamps of the data with one colour for each hour. There is no clear pattern indicating no systematic time of day trend for velocities in either direction. Note how the EXB buoy has larger cross-fjord (north-south) velocity dispersion than does the DXT buoy, compared to the along-fjord dispersionsm, as also indicated by the standard deviation (sd) in the inset of each subplot. As the GPS units are single-channel devices care should be taken in interpreting the different dispersions, and the spectral analysis results in Fig. 13 should be considered at the same time.

be a result of the diurnal cycle. No clear signal was found that could account for this, however. The wind rose plots in Fig. 14 indicate similar persistent directions between 2022 and 2023 across all three termini, mostly aligned with the local topography, suggesting katabatic flow. Tracy and Farquhar glaciers have a more consistently easterly direction, whereas Melville glacier

has a more northerly flow. The wind direction is not different between the two OMB positions however so it is also unlikely to explain any real diurnal cycle.

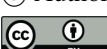


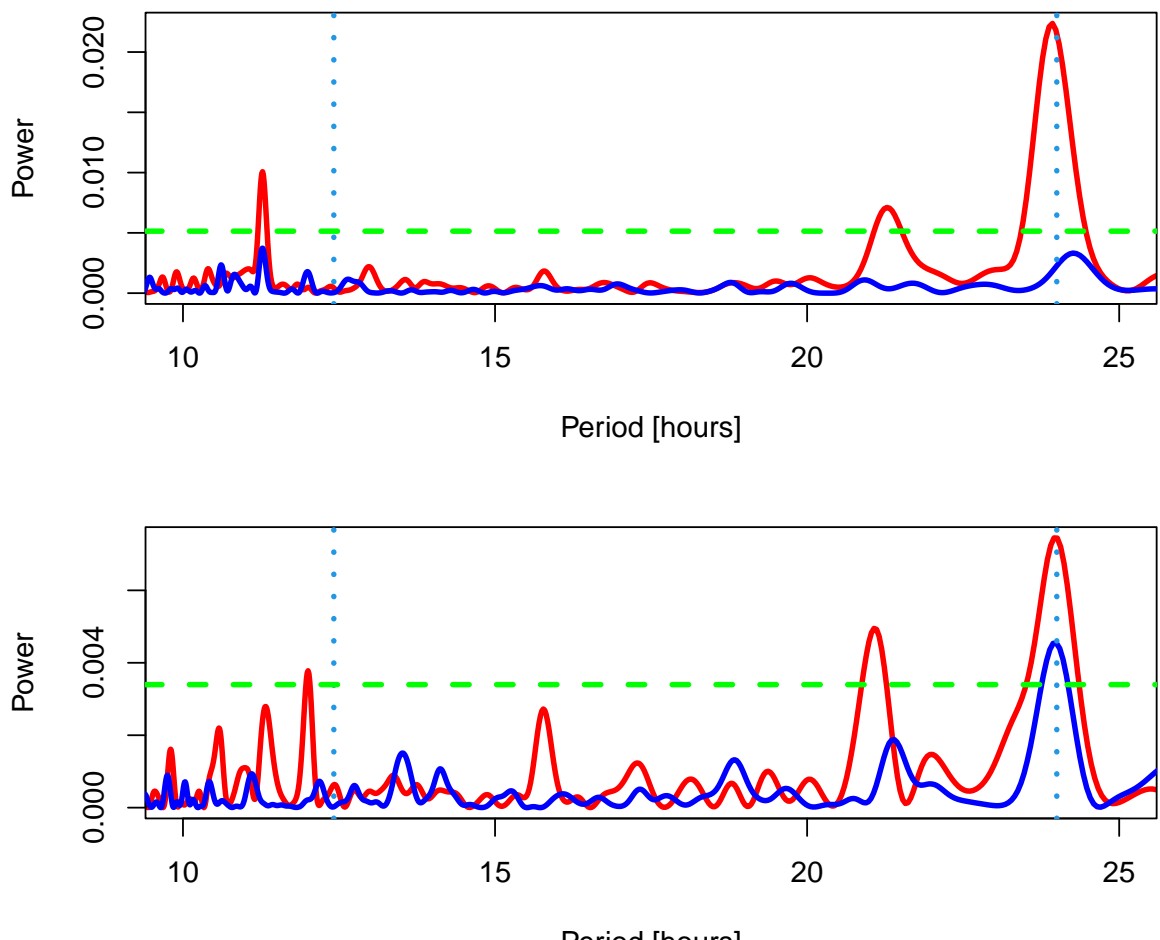

**Figure 13.** Power spectra of OMB buoy speed data - (DXT top, EXB bottom). The intervals were split into first and second halves and the spectra for each half are shown (red is first half, blue last half). The vertical dashed lines indicate the period of the tidal M2 constituent (12 hours 25 minutes) and the 24-hour period. The green dashed line is the significance level at p=0.01. Lomb-Scargle periodogram methods (VanderPlas, 2018) were used because the data are gapped and not evenly spaced.





**Figure 14.** Wind Rose plots from CARRA data for Melville, Farquhar, and Tracy Glacier in 2022 and 2023.

### 3.3.4 Surface mass budget processes

The surface mass budget (SMB) over the glaciers in the region show a classic seasonal cycle in Fig. 15 with winter accumulation followed by ablation driven negative SMB in the summer months. There is no clear diurnal cycle in melt or runoff that could also explain the slight cross fjord pattern of movement in EXB compared to DXT. Figure 15 indicates higher melt and runoff and lower SMB over Tracy glacier compared to the other two, likely reflecting a lower average elevation within the model topography (note that the Fig. 15 shows results only for a single grid point so is comparable in timing for each glacier but overall magnitude differs). The onset of surface runoff coincides with the positive air temperatures in Fig. 10 and the increase in abrupt movements shown in Fig. 3 and 4. We do not see a significant increase in average glacier or mélange velocity in this period but it does coincide with the thinning and eventual break up of sea ice in summer.



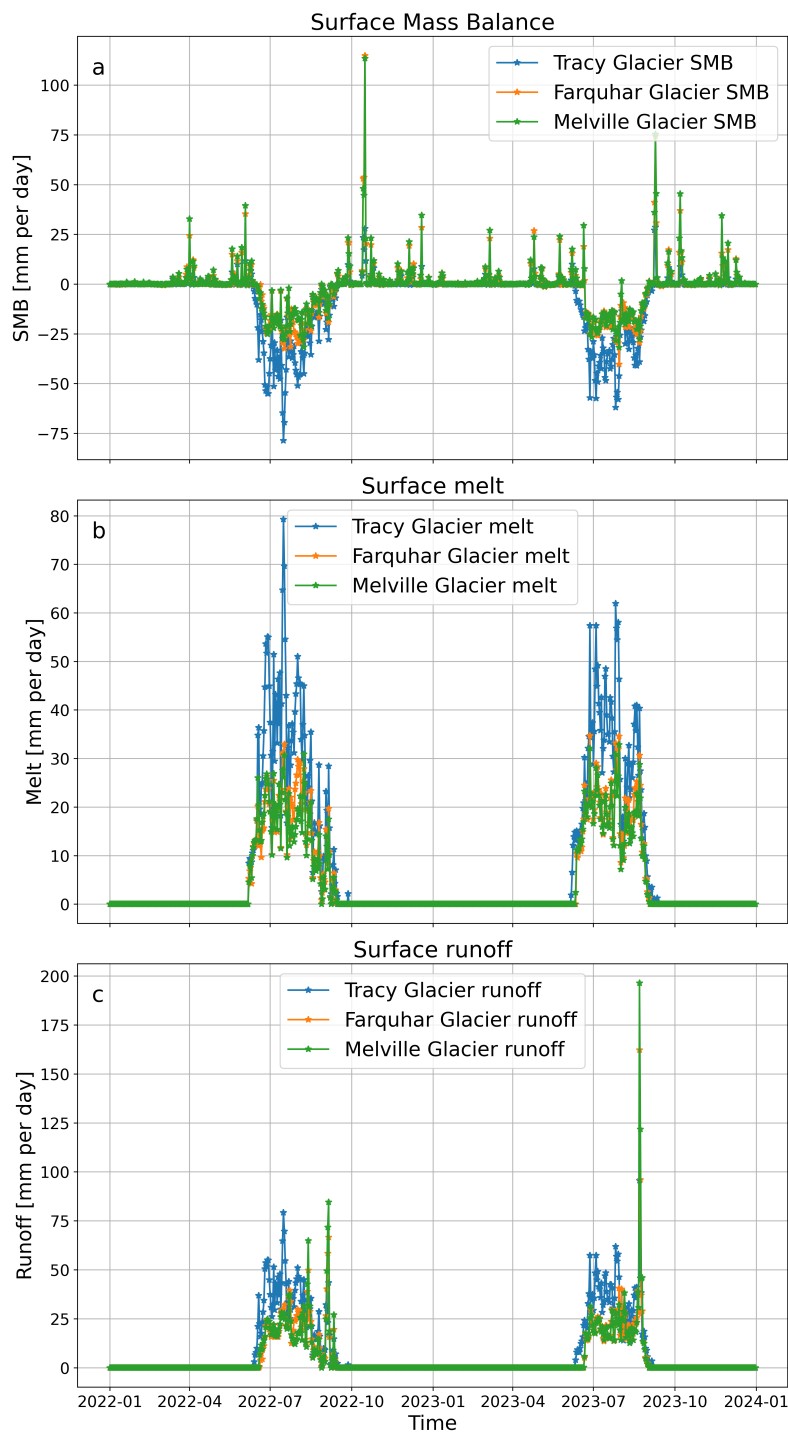

**Figure 15.** Daily SMB (a), surface melt (b), and surface runoff (c) for the three glaciers. All values are given in mm per day, from the model grid cell nearest the glaciers.





## 4 Discussion

Through a comprehensive set of in situ and remote sensing observations, as well as output from regional climate and SMB modelling, we have characterised the mélange zone of three Greenland outlet glaciers through the winter, spring and summer periods. Following the analysis of Bézu and Bartholomaus (2024), we show that the three glaciers demonstrate calving styles

representative of many Greenland calving outlets. Our methods have generated good insights into the development of and deployment of new instruments and techniques in glaciology, as well as highlighting some potential pitfalls and areas for improvement.

### 4.1 Application of new techniques to study mélange processes

The wide availability and near real-time updating of remote sensing data (Solgaard et al., 2021) provided us with an excellent

opportunity to monitor outlet glaciers and derive insights into important cryosphere processes and to evaluate earth observation data products. Machine learning offers an automated means to analyse this data and the application of deep learning to track calving front position is one obvious benefit. We note a few limitations of both manually determined and deep learning extracted calving fronts. While useful for tracking calving fronts on seasonal to annual timescales, the automated calving fronts did not have a precision high enough to identify calving front dynamics on a daily time scale. This is likely due to the fact that the

three glaciers analysed are not part of the training data used for the deep learning model, which reduces the accuracy and consistency of the delineation. In addition, the strong ice mélange and calving activity make these glacier fronts difficult to delineate, even manually. We note however, that the technique shows great promise in reducing labour intensive tasks associated with manually digitising calving fronts and improvement and application of the technique will certainly be beneficial in the future for monitoring and research efforts. Our study also demonstrates the great promise in the deployment of cheap, robust

sensors in collaboration with local communities in Greenland. Our analysis though also highlighted the importance of careful calibration of such sensors to avoid over-interpreting what turned out to be artefacts in the GNSS data from these sensors.

### 4.2 Interpretation of in-situ observations

The conceptual framework and analysis of Robel (2017) is relevant to our interpretation of the results. Field observations confirm the existence of fractures consistent with ploughing through the mélange with a crumple zone of ice rubble and cracks

spreading out more widely from the mélange area itself as shown in the photo in Fig. 16 and in the animations of satellite images in the video supplement.




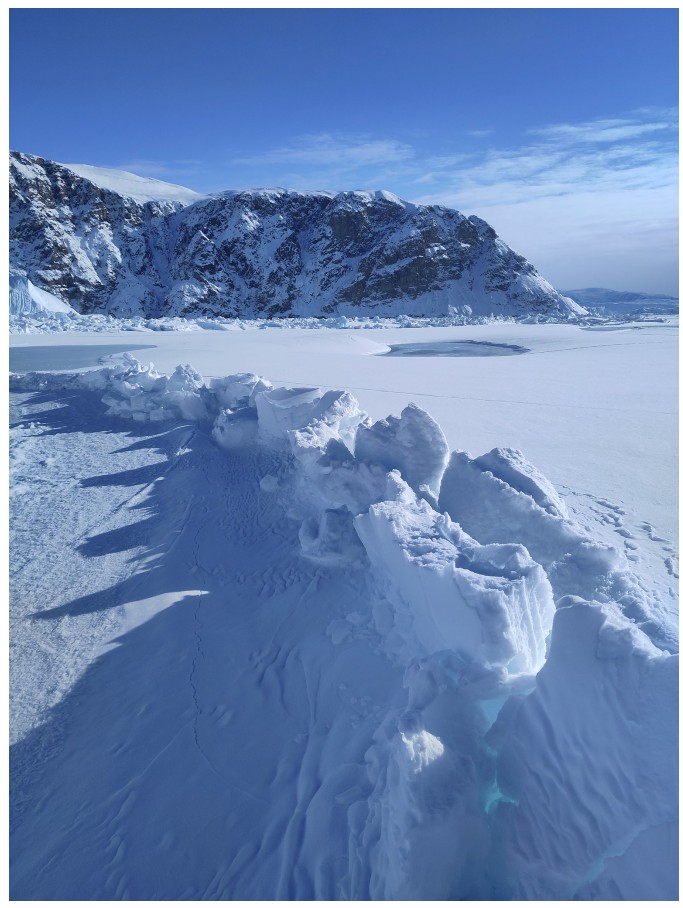

**Figure 16.** The mélange zone in front of Melville glacier with large icebergs visible on the left of the image, in the background the eastern part of the fjord wall is visible. The ridge of ice in the centre is a push ridge indicating the glacier pushing into the landfast ice. Cracks and pools of sea water open under the pressure but quickly freeze shut. The deep snow that covers the sea ice is absent making them easy to pick out.

Robel (2017) suggests that thinning sea ice triggers the onset of the calving activity at the end of the season. Our results do not completely confirm this framework, given that we detect calving activity all through the winter, even when sea ice is at its maximum extent and thickness (see e.g. video supplement too). However, we note that mélange movements do increase
near the end of the season when the sea ice is thin and starts to break up. This movement also coincides with increased calving activity as well as the onset of the melt and runoff season over the land-based glaciers, which also increases glacier velocity (Solgaard et al., 2022). We suggest on this basis that the presence of the landfast ice is more important than the glacier mélange in the winter time in modulating activity, though largely in preventing the removal of calved icebergs. While the landfast sea ice and mélange may exert some back pressure on the calving front, the continuous movement of the mélange away from the
glacier fronts, interrupted by abrupt jumps associated with calving, even during the winter period, suggests that at least in these fjords, the effect of mélange is minimal in controlling ice dynamics to any meaningful extent. The average velocity of the



mélange zone is close to the velocity of the outlet glaciers, particularly Tracy Glacier, through much of the winter period, as measured by the PROMICE velocity data. This suggests a tightly coupled system between glaciers and landfast sea ice, but not one where there is significant back stress exerted. However, the satellite data cannot measure the abrupt movements measured by the GNSS buoys and our analysis shows the importance of very high time resolution and the application of in situ data to accurately measure these events. All buoys show these abrupt events happening synchronously within the time resolution of the instruments. The abrupt movements, corresponding to the release of stress related to fracture of the mélange, occur within the same half hour period. It is difficult to judge therefore if the abrupt movements reflect stress relief in the mélange that then leads to calving, or if the iceberg calving leads to the abrupt jump in the position of the mélange. Even with a higher time resolution, the OMBs do not resolve sufficiently the propagation of sudden fracture events through the mélange. Results from our GNSS buoys show an almost instantaneous response to the same forcing, in spite of their wide (km scale) spacing. Even so, the remote sensing data combined with the buoys do allow us to link the timing of the abrupt events to calving events at the glacier fronts, even during the winter period. Both OMBs register the same shift as the lower resolution TRUSTED buoys, which gives us additional confidence in the interpretation of the movements. The magnitude of the total distance moved decreases very steadily with distance from the glacier fronts as Table A1 in Appendix A1 also makes clear. This observed pattern of movement, suggests that it is calving that is driving the abrupt shifts in mélange position, and not vice versa. The rigidity and coherence of the mélange due to the presence of fast ice is thus likely a more important factor to consider for glacier calving dynamics than the buttressing properties of unfrozen mélange, as also for example suggested by Robel (2017). After the sea ice break up dates, the movement of the mélange buoys is very clearly different and once afloat, they typically left the fjord within a few days and were collected by partners in the local community of Qaanaaq. With the Sentinel-2 satellite imagery (see video supplement) after the landfast ice breaks up, we observe both extensional, compressional and uniform flows of mélange, sometimes simultaneously but usually spatially distributed across the fjord. The difference in the flow regimes before and after sea ice break-up have also been proposed by Burton et al. (2018) and our observations support this analysis.

## 4.3 Controls on Calving Processes

The calving front positions of all glaciers, but especially Melville and Tracy show a similar seasonal timing in both 2022 and 2023 (Fig. 7 & 8), with a net advance from mid-September/start October to mid/late July and a net retreat from mid/late July to mid-September/start October. We note that the pattern in frontal position shown in Fig. 9 is similar to the terminus change time series shown in Figure 1 of Bézu and Bartholomaus (2024) for the three different styles of calving they identify as prevalent in Greenland. Our analysis also therefore supports theirs that although local factors are important for each individual terminus, controls on calving processes can be inferred from general first principles. As all glaciers have similar atmospheric and ocean properties, these appear to be the dominant control on seasonal variability in the front position. The seasonal cycle confirms earlier studies on other glaciers (e.g. Black and Joughin (2023); Moon et al. (2015); Brough et al. (2023)), although Melville, Farquhar, and especially Tracy show a later onset of retreat than other termini in Greenland, possibly related to their relatively northerly position and consequent later break-up of seasonal sea ice. For example, Black and Joughin (2023) showed that the onset of summer retreat patterns occurs later as you move north on the west side of Greenland. Moon et al. (2015)





also concluded that the onset of retreat generally corresponds with break-up of sea ice and onset of higher surface melt, which makes it difficult to distinguish the underlying importance of both factors (Cook et al., 2021; Fried et al., 2018) (see also Fig. 10). Other studies have suggested an increase in glacier velocity could lead to enhanced calving (Amundson et al., 2022; Sakakibara and Sugiyama, 2020) and given the importance of melt and glacier sliding on the velocities in this region it is likely

that these processes are also moderating the calving activity, particularly at Tracy and Melville Glaciers. We examined the relation between sea ice break-up, onset of melt and other possibly driving factors in the overview of calving events displayed in Table A1 in A1. The effect of glacier front instability through changes to the geometry was clear in our manual observations of calving activity. These are calving events where one large event causes the glacier front geometry to be in an unstable configuration, eventually leading to another calving event. An individual calving event may have multiple causal factors, many

of which are coincident in time and space. These factors include ice velocity, terminus topography, bathymetry and basal hydrology as well as local climate and have been identified in other studies (see for example, Amundson et al. (2020); Cowton et al. (2023); Cook et al. (2021); Kneib-Walter et al. (2021); Fried et al. (2018); Benn et al. (2007)). We found no diurnal or tidal influence on mélange velocities or calving events during the fast ice season and do not find a strong diurnal velocity signal in the PROMICE velocities due to the multi day smoothing. However, we do detect a small increase in the glacier velocity in 4

out 10 events detected in 2022. In this context, we can conclude that the presence or absence of mélange at these three termini seems rather unimportant for the calving dynamics. Our results may seem to be apparently at odds with results at other glaciers that suggest an important role for mélange in modulating calving activity. We note however, that studies of mélange dynamics have been overwhelmingly carried out at very large and dynamic outlet glaciers such as Sermeq Kujalleq (Jakobshavn Isbræ) and Helheim Glacier (Joughin et al., 2020; Cassotto et al., 2021; Amundson et al., 2010; Everett et al., 2021). These glaciers

have very large particle sizes in their mélanges, which may promote particle jamming (Peters et al., 2015)) more than in the mélange zones of our study glaciers. There is also a strong summer bias in field studies of glacier mélange for obvious reasons of accessibility. Where other more typical Greenlandic or Alaskan tidewater glaciers have been studied, and particularly in the winter to spring period, their results are more consistent with our findings (Kneib-Walter et al., 2021; Amundson et al., 2020; Black and Joughin, 2023). We note also the results of Wheel et al. (2024), whose three dimensional full stokes model

of Sermeq Kujalleq (Jakobshavn Isbræ) showed the relative unimportance of mélange in controlling calving dynamics and terminus position compared to local geometry.

## 4.4   The role of landfast sea ice

Although our results focus on a single large fjord with multiple glaciers in Northwest Greenland, the geographic situation is much more typical of most Greenland outlet glaciers than the very large outlets typically studied (Bjørk et al., 2015). We

therefore use this analysis to draw conclusions about outlet glaciers in Greenland more widely and, with some reservations to outlet glaciers in Antarctica. Firstly, our results confirm the importance of sea ice in seasonally limiting to some extent, calving from active glaciers. The recent acceleration of Crane glacier on the Antarctic peninsula as well as the seasonal cycle in peninsula glaciers (Surawy-Stepney et al., 2024) have all alternatively been attributed to the loss of rigid multi-year sea ice. Surawy-Stepney et al. (2024) propose, however, that the direct buttressing effect of landfast sea ice on glaciers in the Larsen



B embayment (Antarctica) is minor and that the observed acceleration of glaciers after the disintegration of landfast sea ice is due to secondary processes like increased swell activity in open water, which in turn destabilizes the ice shelves. Alternatively, Surawy-Stepney et al. (2024) postulate that landfast sea ice has a stabilizing effect on the ice mélange in front of the glaciers (for example by inhibiting export of icebergs) and that stable ice mélange in turn slows down the glaciers. The same effect might be at play also in the ice mélange zone we observed in front of Tracy, Farquhar and Melville glaciers, though, perhaps

because the seasonal landfast sea ice is thinner than multi year ice in Antarctica, we see little evidence of mélange slowing ice velocity. This aligns with Robel (2017) who also assess little back stress from landfast ice thinner than 2 m. Similarly, our results do not support the idea that glacier mélange can act as a significant buttress to outlet glaciers in the absence of landfast sea ice during marine ice sheet retreat and/or following the loss of ice shelves as for example proposed by Crawford et al. (2021), though again the thickness of mélange and/or fast ice may play a role as suggested by Robel (2017). We note

the importance of the topographic configuration of our study area. The three glaciers are relatively fast moving, but found in rather narrow fjords with a single outlet to the ocean. They are all in relatively deep water (as measured in the field), where the terminus likely reaches the observed core of warmer Atlantic water at around 200 - 300 m depth. Larger termini in more open embayments may respond differently to the presence of mélange (Meng et al., 2024), though we note that for example the Totten Glacier in Antarctica (Greene et al., 2022) does display a sensitivity to the presence of landfast sea ice in seasonal

velocity patterns. The period of this study is too short to show any trends related to climate or otherwise in the fjord system, however, anecdotal evidence from interviews with local hunters suggests that the fast sea ice season in this fjord system has changed over the last 50 years. Observed sea ice thickness was around 1 m throughout the fjord system in both years. The sea ice is therefore typically much thinner compared to photographs showing almost 2 m thick ice in the 1960s. The fast ice has also formed later and later over the last decade when observations in the fjord monitoring programme run by DMI have

been made in this region. In addition, winter storms that have broken up early season thin sea ice have been noted in the last ten years, while being unknown in the period since settlement in Qaanaaq in the 1950s. These anecdotal results suggest that thinning sea ice over a shorter season may affect the retreat of the outlet glaciers as it allows the mélange to be evacuated from the fjords more quickly. While several models incorporating iceberg mélange exist (Krug et al., 2015; Brough et al., 2023; Everett et al., 2021; Schlemm et al., 2022; Meng et al., 2024), few of them have been tested against a range of datasets or for

general application in Greenland, probably because in situ observations of ice mélange are rare. We therefore also promote the importance of testing models against observational datasets such as this one.

## 5 Conclusions

In this paper we examine the effect of landfast sea ice and iceberg mélange on three representative outlet glaciers in north west Greenland using a range of in situ and remote sensing observations. Our study suggests that while mélange may provide

a short-term and seasonally varying control on outlet glacier fronts, this is largely limited to periods when there is landfast sea ice that hinders the evacuation of calved icebergs. We note a steady drift of mélange away from the glaciers even during winter, with a series of abrupt shifts in position measured by in situ GNSS buoys that are likely due to calving events. As the



sea ice starts to thin and break-up in the summer period, the abrupt movements associated with calving events become more frequent. While this may indicate the importance of the landfast ice in enhancing the rigidity and potential backstress effect of

mélange on the glacier fronts, it is also notable that the summer regime is concurrent with enhanced meltwater and enhanced outlet glacier velocities (Solgaard et al., 2022) and disentangling cause and effect is thus not straightforward. The multiplicity of factors hinders the development of broader parameterisations of the effect of mélange and sea ice on glaciers and ice sheets in ice sheet models, but will be important to capture in order to reproduce glacier fluctuations on seasonal to annual timescales. As climate change promotes earlier onset of melt and sea ice break-up, it will likely enhance rates of calving and ice loss in

Greenland. Our study shows that remote sensing data is very valuable for understanding calving and mélange processes, but notice should be taken of the temporal and spatial resolution of the data. Supplementing with in situ observations is therefore crucial to fully capture all processes at an outlet glacier system. Our datasets will be further used to evaluate and improve the parameterisation of calving and ice mélange in ice sheet models.

*Data availability.*  PROMICE satellite derived velocity data is available on the GEUS dataverse: https://dataverse.geus.dk/dataverse/Ice_

velocity. The automated calving front detection is available here: https://dx.doi.org/10.25532/OPARA-208 CARRA data can be retrieved from the Copernicus climate data store: ttps://cds.climate.copernicus.eu/datasets/reanalysis-carra-single-levels SMB data from HIRHAM5 is available at https://download.dmi.dk/Research_Projects/PROTECT/HIRHAM5_ERA5_GRL/Daily2D_merged/

*Code and data availability.*  The ARCGIS workbenches used to process satellite and in situ observations, together with the associated datasets are archived on zenodo https://zenodo.org/records/15195630:

*Video supplement.*  A full annual calving cycle visualised using Sentinel-2 data is animated in two video supplements on the peertube channel tilvids.com at https://tilvids.com/w/c3TbHzDUu5Lxj5gWd7SgXs and https://tilvids.com/w/jjrNCeQWzMtCnMiaum3wcn and archived on zenodo https://zenodo.org/records/15195630





## Appendix A1 Calving event analysis

Footnotes for table A1

1) Glacier: F - Farquhar, M - Melville, T - Tracy

2) Locations: N - North, S - South, E - East, W - West.

3) Iceberg characteristics: Tu - Turned, Ta - Tabular, CI - Capsized and Integrated in ice mélange.

4) Ice mélange characteristics: R - Rigid, LR - Less Rigid, M - Mixed, O - Open, F - Foggy, Cle - Cleared, D - Dense, LD - Less Dense, CrC - Crack at Corner between Farquhar and Tracy, Cr - Cracks.

5) Tidal cycle: NT - Neap-Tide, ST - Spring-Tide.

6) Glacier geometry: CM - Calving at Margin resulting in geometric instability, CC Calving at Centre as response to geometric instability, SS - more Stable terminus Shape (straight, concave).





**Table A1.** Overview of the 21 analysed calving events and the conditions present around the time of the calving. The table combines relevant results from calving front analysis, satellite and timelapse analysis, detected buoy movements, atmospheric and oceanic conditions from CARRA, and tidal data from Pituffik DMI. Sea ice breakup happened on 16/7/2022 and 25/7/2023. Arrows indicate a gradient in mélange conditions starting at the glacier front. The event number corresponds to Fig. 10.

| No | Date (start) | Date (end) | Glacier[1] | Size of calving [m²][2] | Iceberg type[2,3] | Crevasse | Mélange prior[3,4] | Mélange after[3,4] | Sea ice | Buoy movement | T [°C] | SST [°C] | Wind [m/s] | Tidal[5] | Meltwater plume[2] | Geometry[6] | Other events[1] |
|---|---|---|---|---|---|---|---|---|---|---|---|---|---|---|---|---|---|
| 1 | 6/6/2022 | 11/6/2022 | T | 1344265 | Tu, CI | yes | R, CrC | R | yes | 8/6-9/6 23:30-05:00 Ismaage 804 m, Havoern 650 m | <0 | <0 | 3-4 | | | After: concave | |
| 2 | 16/7/2022 | 17/7/2022 | F | 93985 | CI | | R → M fjord | LR → M | No | | >0 | >0 | 4-6 | | | | M 18/7+19/7 |
| 3 | 17/7/2022 | 18/7/2022 | M | W: 24227 E: 28006 | CI | | Detached and O → LR → O fjord | F | no | | >0 | >0 | 5-6 | | yes W | CM | F 17/7 |
| 4 | 18/7/2022 | 1907/2022 | M | 72801 | E: in notch, CI | | F | LR | no | | >0 | >0 | 3-5 | | | CM | |
| 5 | 19/7/2022 | 25/7/2022 | M | x | Center | | LR | LR, D, Cr → M fjord | no | | >0 | >0 | 2-10 | | Yes, W | CC | |
| 6 | 30/7/22 | 31/7/2022 | M | 22662 | E: in notch, CI | | CIe, except for notch | LR | no | | >0 | >0 | 5 | | Yes, 29/7 | CM | |
| 7 | 9/8/2022 | 16/8/2022 | M | 318933 | Tu, CI | yes | CIe | R LD 2 icebergs moved 600 m | no | | -0 | >0 | 3-11 | | Yes E | After: SS | |
| 8 | 30/8/2022 | 1/9/2022 | F | 71610 | 2 calving events (1 each day), CI | | F | F | no | | <0 | >0 | 5 | | Yes E | | |
| 9 | 12/9/2022 | 13/9/2022 | T | 2185728 | Ta, Tu, CI | | R confined by Tu → LR → O | | no | | <0 | >0 | 4-6, 11/9: 8 | | | After: concave and uneven | M |
| 10 | 12/9/2022 | 13/9/2022 | M | 139608 | CI | | CIe | Expanded, LD, R, 1 Tu moved 2,9 km | no | | <0 | >0 | 4-6, 11/9: 8 | | Yes, E | | T |
| 11 | 20/3/2023 | 21/5/2023 | M | 166208 | CI | | R | R | yes | | <0 | <0 | 5-8 | From NT to ST | | After: concave on each side of an advanced point | |
| 12 | 31/5/2023 | 1/6/2023 | F | | CI | | R, CrC | R, including long narrow iceberg, CrC, iceberg from before moves 160 m | yes | 31/5: Mallemuk 20:00-20:30 80 m | <0 | >0 | 5-6 | NT: 20:20 -833 cm low-tide | | | |
| 13 | 15/6/2023 | 21/6/2023 | M | | Overcast - not visible | | CIe | | yes | 17/6 8:00-8:30 190 m for Soekonge and Havterne | <0 | <0 | 4-8 | From NT to ST or ST | | | |
| 14 | 22/6/2023 | 23/6/2023 | F | 111937 | CI | | R, CrC | R, CrC | yes | 22/6 Mallemuk 6:00-6:30 120 m, 6:20-6:30: OMB EXB 31 m | <0 | <0 | 4-5 | From ST to NT, 132 cm (high-tide) | | | |
| 15 | 16/7/2023 | 17/7/2023 | F | 71165 | W and C, CI | | R, CrC → O | R → LR → M | yes | | >0 | >0 | 4-5 | From NT to ST | | | M |
| 16 | 16/7/2023 | 17/7/2023 | M | 32906 | W, CI | | R → M | LR | yes | | >0 | >0 | 4-5 | From NT to ST | | CM | F |
| 17 | 25/7/2023 | 26/7/2023 | M | 616540 | Ta, CI | Yes | Becoming LR | LR, uniform, Ta from calving moved 1100 m. | No | | >0 | <0 | 4-5 | NT | | Before: imbalance After: concave | F 21/7-25/7 |
| 18 | 30/7/2023 | 09/8/2023 | F | | CI | | R, LD → M | R, LD → LR → M | No | | >0 | >0 | 3-8 | ST | | | |
| 19 | 9/8/2023 | 10/8/2023 | T | 2055345 | Ta size 580652 m², N, CI | Yes | R | LR, expanded | No | no | -0 | >0 | 5 | NT | | After: CM | |
| 20 | 17/8/2023 | 18/8/2023 | T | 3376239 | 4 Ta, 2 Tu, CI | Yes | LR expanded | R confined by iceberg and glacier → LR → O, Ta from 10/8 moves 2,2 km | no | no | -0 | >0 | 5-6 | ST | | Before: Imbalance After: more concave but imbalance | |
| 21 | 6/9/2023 | 7/9/2023 | T | 691568 | Ta, S, N, CI | | R confined by iceberg and glacier→ LR, M, large extend decreased a bit | R confined by iceberg and glacier → LR → M → O | no | no | <0 | >0 | 5 | NT | | After: balance in shape | |



*Author contributions.* This study was conceived and planned and fieldwork was executed by RM, AG and SMO. Analysis of in situ and satellite data was carried out by SH, OBB and RM. SMO provided and analysed ocean data. PT assisted with interpretation of the GNSS and

CARRA data. SMB and CARRA data was provided by MO and NH, the automated calving dataset was provided and analysed by EL and interpretation of ice sheet satellite velocities was provided by AS. The paper was planned by RM, SH and OBB. All authors participated in the writing of the paper

*Competing interests.* Co-author RM is an editor of The Cryosphere. There are no other competing interests.

*Acknowledgements.* We pay tribute to the local community for all their help and hospitality, particularly the hunters and fishers of Qaanaaq,

especially Peter and Qillaq Danielsen and Gustav Simigaq who assisted in transporting us to the field locations and in retrieving the buoys during the summer. Invaluable logistical support in the field was also provided by our DMI colleague Aksel Ascanius.

We acknowledge funding and support of the Danish government in the National Centre for Climate Research (NCKF). We thank Jean Rabault for answering all our questions regarding the OMB buoys and for his kind help with programming them tailored to our needs.

NH was supported by the Novo Nordisk Foundation Challenge project PRECISE (NNF23OC0081251).

MO received funding from the European Union's Horizon 2020 research and innovation programme under grant agreement 869304. This is PROTECT contribution number: XX

RM acknowledges the support of the PolarRES project funded by the European Union's Horizon 2020 research and innovation programme call H2020-LC-CLA-2018-2019-2020 under grant agreements number 101003590. We also acknowledge the support of the European Space Agency through the Ice Sheets CCI (Grant 4000126523/18/I-NB) for development and application of the remote sensing datasets and useful

discussions.



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
