# Peer review of "Mélange or landfast ice: What controls seasonal calving at Greenland outlet glaciers?"

_EGUsphere, 2025_

## Referee Comment (RC2)

**Overview**

This study makes use of a variety of datasets collected in the field and supplemented by satellite derived remote sensing data to investigate the role that landfast sea ice and melange may play in regulating iceberg calving at three outlet glaciers in Northwest Greenland.

Whilst the data collected and analysed in this study is novel and provides a fantastic opportunity to add nuance to the dynamic behaviour of the combined glacier, melange and fast ice regions over much shorter time intervals than is possible using satellite data, the evidence as currently presented is not a robust proof of the conclusions drawn from the study. Further use of remote sensing data could be used to investigate correlations between the extent and thickness of landfast sea ice and to gain further insight into the overall calving dynamics of the glaciers. The role of melange and fast ice in buttressing and suppression of calving is an important topic to address and with major revisions to the manuscript, this study could be very beneficial to furthering understanding in this area.

**Major Comments**

- It is not explicitly stated in the methods how the roles of landfast sea ice and melange are distinguished in modulating the calving behaviours. This appears to be given by the extent of the fast ice? Further clarification could be given on this i.e. how the spatial coverage of the fast ice within the fjord changes throughout the assessed time period and how this correlates to calving and velocity changes. Satellite imagery could be used to quantify this. Similarly, the thickness of the fast ice could be an important factor – are any measurements available (either from the field or from altimetry) to quantify how the thickness changes over the time period? Note that times of maximum and minimum sea ice thickness are mentioned in the manuscript but it is not clear how these thicknesses were measured. Again, it would be interesting to see how thickness correlates to the velocities and calving dynamics.

- The distinction between velocities at different points within the glacier compared to velocities within the fast ice may be misleading. We may expect the velocities to increase as the ice approaches the terminus (or flotation – are they grounded at the time of these measurements as they are mentioned to transiently float in the introduction?). A 2D velocity map would show the spatial change in velocity better and could still be used to illustrate the temporal changes in velocity as Figure 5b attempts.

- The differences in velocities as presented in Figure 5b are used to infer that the fast ice does not significantly buttress the glaciers (Section 4.2). The velocities can be used to calculate strain rates and with knowledge of the thickness of the fast ice the backstress can be quantified, so we don't need to speculate on this.

- The conclusions of the study are based on calving events identified through satellite imagery (shown in Figure 10). Whilst larger calving events may be discernible from satellite imagery, smaller events cannot be identified, as is discussed in the manuscript. Assessing calving through this methodology may therefore not tell the whole story. The 'overall' calving rate could be analysed from remote sensing data as the difference between the glacier's flow velocity and the change in the calving front position over time. Considering averaged calving rates in this way rather than as discreet events may shed more light, in particular if they can be plotted against changes in fast ice thickness / extent.

- It is difficult to conclude from the results as currently presented where correlation and causation between the factors discussed in the study lie. Restructuring the results could therefore be beneficial in comparing the relationship between all of the variables discussed. For example could a single plot show a time series of velocities, calving rates, sea/air temperatures, smb/melt rates etc. would make a nice visual comparison of correlated events.

- The glaciers assessed in this study are discussed as being widely representative of Greenland's outlet glaciers due to the different calving styles observed however further information on the flow speeds / widths / thicknesses etc of the glaciers may help the reader to determine how representative these glaciers are in the wider context. Adding this information in a table may help.

We also need to understand the thickness and the extent of the fast ice and melange regions as these may vary in different regions around Greenland. Similarly, the topography/confinement of the fjord may also impact how readily backstresses from fast ice may be transmitted to the glaciers. It is therefore not clear whether the results presented here can be considered representative across Greenland.

**Specific Comments**

Figure 2 – It is later discussed that the start points of the buoys were influenced by accessibility / patterns of melange then a basemap from around the time of deployment would be useful. Were certain spatial coverages of the buoys targeted?

Line 227 – 'the melange velocities when bounded by landfast sea ice' - when exactly were these not bound by the fast ice? Can a figure be referenced in relation to this statement?

Line 236 – 'a more spatially dispersed dataset is necessary to fully determine this' – can satellite data be used in this sense?

General – the results sections tend to contain some interpretation which may be better placed in the discussion sections - and in some cases is repeated/contradicted later in the discussion. For example line 241 – 'these large abrupt movements likely represent large calving events' and line 408/409 'it is difficult to judge therefore if the abrupt movements reflect stress relief in the melange that then leads to calving, or if the iceberg calving leads to the abrupt jump..'

Figure 3 – the plot scale on panel b) could be changed so we can better see the velocities for the majority of the data – which are mostly <50 m d$^{-1}$ (note units don't need the 'x' in m x d$^{-1}$). In panel a) it is not clear why a base map from 2024 is used? Could large iceberg fragments influence the lateral deviation in movement (particularly seen in red / blue / orange / yellow buoys? There may be a better visual representation of this. The grey outline of the start point is not clear. Same comments apply to Figure 4.

Line 263/264 – 'The Sentinel-1 data has a 12 day repeat orbit which means it cannot be used to assess the abrupt events identified in the GNSS buoy data' . Do the calving front time series in Figure 9 not correlate? Plotting the calving front time series under fig 3b / 4b may allow correlation to be identified?

Line 268/269 – the melange zone and outer fast ice zones are mentioned a few times – can you mark on a figure what you mean by these zones?

Figure 5 – Flow speeds are shown, not velocities. Can 2D velocity maps be used to better illustrate the spatial changes in velocity?

Figure 6 / 7/ 8 – The colour scheme makes it quite difficult to follow the difference between sequential months. Are the spatial scales different? If so why?

Figure 7 /8 – is it necessary to plot the calving fronts which are clearly wrong and can be discounted from the datasets? Are any higher resolution basemaps available?

Figure 9 – can the central flow lines that these refer to be marked on Figures 6 -8 for reference? More subdivisions on the x axis would make it easier to see how this correlates to the displacements shown in figs 3 and 4. Even better to combine with figs 3 and 4.

Figure 10 – title not required – this is covered in the caption.

Figure 12 – Refer to panels a and b (rather than left and right) for consistency. Titles are not needed and they reference Qaanaaq – should this be a specific outlet glacier? Can a colour scale or legend be added to show what times the dots refer to?

Figure 13 - What are the intervals that are referred to in the caption? The caption could be a bit more detailed to ensure that the figure can be interpreted without needing to refer back to the main text. Refer to panels a and b (rather than top and bottom) for consistency.

Line 439 – 'given the importance of melt and glacier sliding on velocities, it is likely that these processes are also moderating calving activity'. It was discussed in Section 3.3.4 that a significant increase in glacier / melange velocities were not observed in correlation with runoff, please can you clarify this? Plotting the panels in Fig 15 against velocities and calving rates could better display correlation or lack of.

Line 449/450 – 'we do detect a small increase in the glacier velocity in 4 out *of* 10 **calving** events detected in 2022' – can this be shown by plotting calving vs velocity? Line 450/451 goes on to say 'we can conclude that the presence or absence of melange … seems rather unimportant for calving dynamics'. This conclusion does not seem robust to me. It could be that these are large tabular style caving events in which a large amount of energy is released – in which case we may not expect backstresses from fast ice / melange to be of a magnitude large enough to suppress this. The melange may have more impact on suppressing smaller calving events (which could be analysed by assessing overall averaged calving rates), which in turn could moderate the glacier velocities – in which case it could be argued that the melange does play an important role. Neither conclusion is clear to me on the evidence provided so far, but a plot of melange / fast ice thickness and extent against velocities and calving rates could go a long way to showing this. I wonder also if discreet calving events could be looked at separately to overall calving rates – i.e. does the calving type matter and does melange / fast ice inhibit the smaller calving events but have little impact on the larger scale events?

**Typos**

Line 44 – 'spring an**d** early summer' ?

Line 99 – '… all outlet glaciers i**n**s not well constrained'

Line 312 – 'observations **of** ocean processes'

Figure 12 caption – 'dispersions**m**'

---

## Author Comment (AC3)

Review 2

Overview

This study makes use of a variety of datasets collected in the field and supplemented by satellite derived remote sensing data to investigate the role that landfast sea ice and melange may play in regulating iceberg calving at three outlet glaciers in Northwest Greenland.

Whilst the data collected and analysed in this study is novel and provides a fantastic opportunity to add nuance to the dynamic behaviour of the combined glacier, melange and fast ice regions over much shorter time intervals than is possible using satellite data, the evidence as currently presented is not a robust proof of the conclusions drawn from the study. Further use of remote sensing data could be used to investigate correlations between the extent and thickness of landfast sea ice and to gain further insight into the overall calving dynamics of the glaciers. The role of melange and fast ice in buttressing and suppression of calving is an important topic to address and with major revisions to the manuscript, this study could be very beneficial to furthering understanding in this area.

We thank the reviewer for recognising the contributions we hope to make with this study. We also appreciate the suggestions for widening the Earth Observation part of the study, but we do not intend to make this a remote sensing study but would rather like the in situ observations to take centre stage. We have included the remote sensing analysis to help make sense of some of the in situ measurements and we think this allows a clean interpretation of the measurements. We are very happy to collaborate with remote sensing groups who would like to use our data for calibration, as well as for ice sheet/mélange modelling and have made this clearer in the paper too.

Major Comments

• It is not explicitly stated in the methods how the roles of landfast sea ice and melange are distinguished in modulating the calving behaviours. This appears to be given by the extent of the fast ice? Further clarification could be given on this i.e. how the spatial coverage of the fast ice within the fjord changes throughout the assessed time period and how this correlates to calving and velocity changes. Satellite imagery could be used to quantify this.

R1 had a similar comment and we have clarified how we interpret landfast ice and mélange as a continuum in the abstract and in section 1.1. We did indeed use satellite imagery to identify the break up of the landfast ice in the fjord and we have added a few more details in the methods describing this. The animations of satellite images linked to in the supplement are also referred to on this point. The break up of the ice in the mélange zone is also clearly visible in the GPS buoy records, and very soon afterwards all are circulated out of the mélange zone and collected by our local collaborators in the region, for which reason we stop the tracking when the buoys start to float.

Similarly, the thickness of the fast ice could be an important factor – are any measurements available (either from the field or from altimetry) to quantify how the thickness changes over the time period? Note that times of maximum and minimum sea ice thickness are mentioned in the manuscript but it is not clear how these thicknesses were measured. Again, it would be interesting to see how thickness correlates to the velocities and calving dynamics.

Fast ice thickness was measured at all sites during deployment as well as at CTD deployment sites and as part of observations of opportunity and was found to be remarkably consistent in thickness all through the fjord, but we do not have continuous measurements near the glacier termini following deployment. Altimetry is unfortunately not an easy proposition so close to the coasts and in the narrow fjords and we found that even high resolution datasets such as ArcticDEM, TerraSARX etc do not represent the mélange zone at sufficient resolution. We are in the process of compiling a detailed sea ice thickness database from direct measurements in the region in a data rescue project, to be published in 2026, which will hopefully also be a useful dataset for future studies. We have noted this in the discussion section.

The distinction between velocities at different points within the glacier compared to velocities within the fast ice may be misleading. We may expect the velocities to increase as the ice approaches the terminus (or flotation – are they grounded at the time of these measurements as they are mentioned to transiently float in the introduction?). A 2D velocity map would show the spatial change in velocity better and could still be used to illustrate the temporal changes in velocity as Figure 5b attempts.

We have plotted the 2D velocities (see below) and will add them as an appendix to the paper with some additional explanation in the results section as a supplement to the current point velocities shown in the main manuscript to illustrate our general findings about the mélange velocities. They show very much the expected pattern with slower margins along the fjord side and a faster centre line and also increasing velocity close to the calving front.

It is unclear how much of each glacier termini is floating or when, but we interpret the occurrence of tabular icebergs as a marker of at least partial transient flotation close to the terminus. We have added some additional text to the paper on this point as well as including the following additional figure.

[Figure]

The differences in velocities as presented in Figure 5b are used to infer that the fast ice does not significantly buttress the glaciers (Section 4.2). The velocities can be used to calculate strain rates and with knowledge of the thickness of the fast ice the backstress can be quantified, so we don't need to speculate on this.

We have calculated the first principal strain rate based on the satellite derived velocities (see below), and we will also add this to the supplementary materials. Unfortunately, the satellite data is rather noisy at the calving front and in the mélange zone, and this makes it complex to interpret beyond the glacier front. In addition, the thickness of the fast ice and mélange is also difficult to reduce to a single number or even grid of numbers as it is a very heterogenous material depending on how large the icebergs are in between the fast ice which also becomes ridged and crumpled through the winter as the glaciers advance into it. A full stress and strain analysis of the region is beyond the scope of this study but is the focus of a follow-on project where a UAV was used to map the surface of parts of the mélange area during buoy deployment. This is being processed to derive a particle size distribution of icebergs and ice floes and could be incorporated into a mélange model.

[Figure]

The conclusions of the study are based on calving events identified through satellite imagery (shown in Figure 10). Whilst larger calving events may be discernible from satellite imagery, smaller events cannot be identified, as is discussed in the manuscript. Assessing calving through this methodology may therefore not tell the whole story. The 'overall' calving rate could be analysed from remote sensing data as the difference between the glacier's flow velocity and the change in the calving front position over time. Considering averaged calving rates in this way rather than as discreet events may shed more light, in particular if they can be plotted against changes in fast ice thickness / extent.

In the manuscript we present the calving dynamics at the three glaciers in several different ways, including the overall frontal advance and retreat through time in Figure 9 as well as the monthly terminus position in Figures 6,7 and 8 and 10. The suggestion to calculate an overall "calving rate" is a good one but it is already covered by the analysis we show here, especially given the adjustments we have made to Figure 9 with some additional modifications showing the breakup of fast ice and the days with positive temperatures. The analysis in the table in the appendix, which Figure 10 is based on, is an attempt to identify large calving events which may have an external cause such as for example the break up of sea ice or onset of high surface melt. Changes in fast ice thickness are hard to identify as we do not have a continuous observation of this and the extent in the fjord typically follows a rather abrupt pattern. As we observe in the discussion section, many different processes that may affect calving peak at around the same time, so deriving causation from correlation is very difficult and several processes are likely co-causal (see also below). We have added emphasis on this point in the discussion section which has also been restructured to discuss the different processes more clearly.

It is difficult to conclude from the results as currently presented where correlation and causation between the factors discussed in the study lie. Restructuring the results could therefore be beneficial in comparing the relationship between all of the variables discussed. For example could a single plot show a time series of velocities, calving rates, sea/air temperatures, smb/melt rates etc. would make a nice visual comparison of correlated events.

We agree that it is difficult to disentangle causation and correlation as many of the factors are deeply entwined and in fact are also co-causal. We have therefore restructured the discussion section as requested to draw out the different processes and we will also clarify this point further, including drawing on the following changes to the figures, as also requested by R1; a) we will add the sea ice break up date lines to Figures 9 and 15; b) we have added the same positive and negative colours shown in Figure 10 to the calving front position plots in Figure 9. We believe this gives a greater clarity and harmony to all figures rather than trying to compress all processes into one single figure.

The glaciers assessed in this study are discussed as being widely representative of Greenland's outlet glaciers due to the different calving styles observed however further information on the flow speeds / widths / thicknesses etc of the glaciers may help the reader to determine how representative these glaciers are in the wider context. Adding this information in a table may help. We also need to understand the thickness and the extent of the fast ice and melange regions as these may vary in different regions around Greenland. Similarly, the topography/confinement of the fjord may also impact how readily backstresses from fast ice may be transmitted to the glaciers. It is therefore not clear whether the results presented here can be considered representative across Greenland.

We will add the details of glacier and fjord geometry to the description of the study sites, though note that we are unsure how reliable the standard outlet glacier thickness data from e.g. Bedmachine or other basal datasets in this region actually is. The style of calving discussed in the manuscript and summarised in the table in the supplement is very much aligned with the analysis of Bezu and Bartholomeus, 2024 as we point out in the manuscript. The seasonal cycles in velocity also align with those analysed using K-means clustering by Solgaard et al 2023 and we therefore consider these outlet glaciers very representative of the small to medium-sized glaciers but likely not the very large outlets like Jakobshavn and Helheim Glaciers.

Specific Comments

Figure 2 – It is later discussed that the start points of the buoys were influenced by accessibility / patterns of melange then a basemap from around the time of deployment would be useful.

Were certain spatial coverages of the buoys targeted?

It is hard to get a good basemap of the dates of deployment as the relatively early date of deployment means optical images suffer from shadows and or cloud cover that obscure the mélange. In fact, while we do use satellite imagery to plan deployment, actual conditions are verydifficult to interpret from the pictures and often different than imagined on the ice. It is rarely as easy to access particular locations as the imagery can suggest. This is difficult to see on satellite images as the size of the icebergs and ice floes is not easily captured at relevant spatial scales. We therefore prefer to use a single standard basemap for both seasons.

Line 227 – 'the melange velocities when bounded by landfast sea ice' - when exactly were these not bound by the fast ice? Can a figure be referenced in relation to this statement?

Apologies, this is a little ambiguous, all of the position data we show here is when the mélange is locked up by landfast ice as the buoys circulate out of the mélange zone very quickly after the sea ice breaks up. We will add this to the methods sections.

Line 236 – 'a more spatially dispersed dataset is necessary to fully determine this' – can satellite data be used in this sense?

While satellite data could certainly help, the point about our method is to capture the very brief high time resolution movements, which satellite data cannot capture. However, we hope to continue these deployments in future years to get a longer time series of data. We have clarified this statement as we intended to mean more deployments at different locations in front of the glaciers.

General – the results sections tend to contain some interpretation which may be better placed in the discussion sections - and in some cases is repeated/contradicted later in the discussion.

For example line 241 – 'these large abrupt movements likely represent large calving events' and line 408/409 'it is difficult to judge therefore if the abrupt movements reflect stress relief in the melange that then leads to calving, or if the iceberg calving leads to the abrupt jump..'

Thank you for this feedback, we have edited and slightly restructured both results and discussion sections to avoid mixing the two and make the interpretation more clear.

Figure 3 – the plot scale on panel b) could be changed so we can better see the velocities for the majority of the data – which are mostly <50 m d-1 (note units don't need the 'x' in m x d-1). In panel a) it is not clear why a base map from 2024 is used? Could large iceberg fragments influence the lateral deviation in movement (particularly seen in red / blue / orange / yellow buoys? There may be a better visual representation of this. The grey outline of the start point is not clear. Same comments apply to Figure 4.

Thank you for the suggestion we have updated the figures 3b and 4b to show a log scale and also enlarged the dots to make the figures clearer, see also our reply to the basemap request above. We discuss the role of the larger icebergs in the discussion too.

Line 263/264 – 'The Sentinel-1 data has a 12 day repeat orbit which means it cannot be used to assess the abrupt events identified in the GNSS buoy data' . Do the calving front time series in Figure 9 not correlate? Plotting the calving front time series under fig 3b / 4b may allow correlation to be identified?

The table in the supplement are intended to try and correlate calving activity with the buoy movements shown in figures 3b and 4b, based on visual identification of calving front changes. We have clarified this in the text.

Line 268/269 – the melange zone and outer fast ice zones are mentioned a few times – can you mark on a figure what you mean by these zones?

Yes we will add these labels to Figure 2.

Figure 5 – Flow speeds are shown, not velocities. Can 2D velocity maps be used to better illustrate the spatial changes in velocity?

We have plotted up a 2D velocity map and will put it into the supplementary material as it does not substantially change our conclusions.

Figure 6 / 7/ 8 – The colour scheme makes it quite difficult to follow the difference between sequential months. Are the spatial scales different? If so why?

Figures 6-8 are a continuous dataset and the colours are chosen to be sequential from white through light to dark blue showing how the front varies through time. We will update this in the caption. However, we also discovered a slightly offset map frame, which made the scales appear different. We have corrected this so they all appear the same.

Figure 7 /8 – is it necessary to plot the calving fronts which are clearly wrong and can be discounted from the datasets? Are any higher resolution basemaps available?

There are no higher resolution basemaps unfortunately as these are rather small glaciers, and we feel that while some of the calving fronts are very likely wrong in the plot it's a nice illustration of the point we make in the discussion that while automated tracking algorithms based on machine learning methods are really useful, they still require some work and need to be checked by a human user. We have made this point clear in the caption too.

Figure 9 – can the central flow lines that these refer to be marked on Figures 6 -8 for reference? More subdivisions on the x axis would make it easier to see how this correlates to the displacements shown in figs 3 and 4. Even better to combine with figs 3 and 4.

This is a nice suggestion which we tried out, but unfortunately, as the scaling is different between the two plots, it made them rather hard to interpret. We have therefore removed it again. We prefer to keep both plots as they are since it illustrates the difficulty in representing a three dimensional object through time in a simple figure.

Figure 10 – title not required – this is covered in the caption.

This has been fixed

Figure 12 – Refer to panels a and b (rather than left and right) for consistency. Titles are not needed and they reference Qaanaaq – should this be a specific outlet glacier? Can a colour scale or legend be added to show what times the dots refer to?

This has been fixed. The title was indeed unnecessary and we prefer not to refer to a specific glacier as the position of the buoys was between two of the three. The colours refer to specific times in the day, but not to specific days. We have updated the caption to reflect this and will add a colour scale too.

Figure 13 - What are the intervals that are referred to in the caption? The caption could be a bit more detailed to ensure that the figure can be interpreted without needing to refer back to the main text. Refer to panels a and b (rather than top and bottom) for consistency.

Thanks for this feedback we have updated the caption to give more detailed information on interpreting the spectral analysis. The intervals referred to are simply the first half of the record and the second, to assess if there is any change over the period. We have also added a and b to the panels

Line 439 – 'given the importance of melt and glacier sliding on velocities, it is likely that these processes are also moderating calving activity'. It was discussed in Section 3.3.4 that a significant increase in glacier / melange velocities were not observed in correlation with runoff, please can you clarify this? Plotting the panels in Fig 15 against velocities and calving rates could better display correlation or lack of.

Our point here is that many different processes are acting on glaciers and on the fast ice and mélange at the same time. We have updated Figure 9 to show the break up of the fast ice and above/below 0 mean daily air temperatures. Fig. 15 now also marks the point where the land fast sea ice breaks up as identified from the satellite record as a continuous marker point through the plots. We have restructured the discussion to be clearer around how we interpret the different factors that affect the calving processes at the glaciers and clarified in section 3.3.4 that while we don't see an increase in average velocity in the mélange, during this period, we do see an increase in the number of abrupt movements in the mélange record.

Line 449/450 – 'we do detect a small increase in the glacier velocity in 4 out of 10 calving events detected in 2022' – can this be shown by plotting calving vs velocity? Line 450/451 goes on to say 'we can conclude that the presence or absence of melange … seems rather unimportant for calving dynamics'. This conclusion does not seem robust to me. It could be that these are large tabular style caving events in which a large amount of energy is released – in which case we may not expect backstresses from fast ice / melange to be of a magnitude large enough to suppress this. The melange may have more impact on suppressing smaller calving events (which could be analysed by assessing overall averaged calving rates), which in turn could moderate the glacier velocities – in which case it could be argued that the melange does play an important role. Neither conclusion is clear to me on the evidence provided so far, but a plot of melange / fast ice thickness and extent against velocities and calving rates could go a long way to showing this. I wonder also if discreet calving events could be looked at separately to overall calving rates – i.e. does the calving type matter and does melange / fast ice inhibit the smaller calving events but have little impact on the larger scale events?

We have restructured the discussion section fully to a more logical structure following the different processes important for calving processes and frontal position as well as the mélange cycle. Figure 10 together with Table A1 show the relative size of individual calving events identified from satellite imagery and in the GNSS record and we refer to this in this section again. Calving activity occurs all through the winter period (as also shown in the satellite data animation) including both small and large events. The presence of land fast sea ice suppresses the removal of calved icebergs and allows the mélange to build up further, but while it may exert some back pressure on the calving fronts (see also the additional figures showing the principal strain rates), this is not large compared to the forward motion of the outlet glacier. We do also note that this may not be the case at the very large outlets like Sermeq Kujalleq/Jakobshavn Isbræ. As noted before, we do not unfortunately have a good dataset for fast ice/melange thickness at these glaciers yet. However, as we also try to make clear, we focus very much on seasonal calving dynamics in this study.  The position of the calving front does have a clear seasonal cycle at all fronts, likely related to a complex combination of changes in ice velocity, runoff and air temperature and the presence or absence of land fast sea ice as well as to the general stability of the position of the calving front and the dominant calving styles of the glacier, which is also related to bed topography and fjord geometry as well as mass budget and hydrology. We highlight these factors in the restructured discussion section.

Typos –

Thank you for catching these. The following have all now been fixed in the manuscript.

Line 44 – 'spring and early summer' ?

Line 99 – '… all outlet glaciers ins not well constrained'

Line 312 – 'observations of ocean processes'

Figure 12 caption – 'dispersionsm'